# Dynamically linking influenza virus infection kinetics, lung injury, inflammation, and disease severity

Margaret A Myers[1†], Amanda P Smith[1†], Lindey C Lane[1], David J Moquin[2], Rosemary Aogo[1], Stacie Woolard[3], Paul Thomas[4], Peter Vogel[5], Amber M Smith[1]*

[1]Department of Pediatrics, University of Tennessee Health Science Center, Memphis, United States; [2]Department of Anesthesiology, Washington University School of Medicine, St. Louis, United States; [3]Flow Cytometry Core, St. Jude Children's Research Hospital, Memphis, United States; [4]Department of Immunology, St. Jude Children's Research Hospital, Memphis, United States; [5]Department of Pathology, St. Jude Children's Research Hospital, Memphis, United States

**Abstract** Influenza viruses cause a significant amount of morbidity and mortality. Understanding host immune control efficacy and how different factors influence lung injury and disease severity are critical. We established and validated dynamical connections between viral loads, infected cells, CD8+ T cells, lung injury, inflammation, and disease severity using an integrative mathematical model-experiment exchange. Our results showed that the dynamics of inflammation and virus-inflicted lung injury are distinct and nonlinearly related to disease severity, and that these two pathologic measurements can be independently predicted using the model-derived infected cell dynamics. Our findings further indicated that the relative CD8+ T cell dynamics paralleled the percent of the lung that had resolved with the rate of CD8+ T cell-mediated clearance rapidly accelerating by over 48,000 times in 2 days. This complimented our analyses showing a negative correlation between the efficacy of innate and adaptive immune-mediated infected cell clearance, and that infection duration was driven by CD8+ T cell magnitude rather than efficacy and could be significantly prolonged if the ratio of CD8+ T cells to infected cells was sufficiently low. These links between important pathogen kinetics and host pathology enhance our ability to forecast disease progression, potential complications, and therapeutic efficacy.

*For correspondence: amber.smith@uthsc.edu

[†]These authors contributed equally to this work

Competing interests: The authors declare that no competing interests exist.

## Introduction

Over 15 million respiratory infections and 200,000 hospitalizations result from influenza A viruses (IAVs) each year (*Thompson et al., 2004*; *Simonsen et al., 2000*; *Taubenberger and Morens, 2008*; *Medina and García-Sastre, 2011*). The incidence and severity of IAV infections increases when new strains emerge and/or when there is a lack of prior immunity. A robust immune response is crucial for resolving viral infections, but immune-mediated pathology can exacerbate disease (*Duan and Thomas, 2016*; *Moskophidis and Kioussis, 1998*; *Mauad et al., 2010*; *Rygiel et al., 2009*; *La Gruta et al., 2007*). High viral loads can also play a role in disease progression (*Boon et al., 2011*; *de Jong et al., 2006*), but these do not always correlate with the strength of the host response or with disease severity (*Granados et al., 2017*; *Marathe et al., 2016*; *Toapanta and Ross, 2009*; *Smith et al., 2019*; *Gao et al., 2013*). An understanding of how viral loads, host immune responses, and disease progression are related is critical to identify disease-specific markers that may help predict hospitalization or other complications.

During IAV infection in both humans and animals, viral loads increase rapidly for the first 1–2 days of infection before reaching a peak (e.g., as in *Smith et al., 2018*; *Baccam et al., 2006*; *Miao et al., 2010*; *Toapanta and Ross, 2009*; *Carrat et al., 2008*; *Srivastava et al., 2009*; *Gao et al., 2013*). If the host is infected with a novel strain or has no prior immunity, viral loads in the lung then begin to decline, first slowly (sometimes <1 $\log_{10}$ TCID$_{50}$/d ) then rapidly (> 4-5 $\log_{10}$ TCID$_{50}$/d) (*Smith et al., 2018*; *Gao et al., 2013*). We previously quantified this biphasic viral decline with a mathematical model, which indicated that the rate of infected cell clearance increases as the density of infected cells decreases (*Smith et al., 2018*). The timing of the second, rapid viral decay phase coincides with the expansion of CD8$^+$ T cells, which are the primary cell responsible for clearing infected cells and resolving the infection (*Zhang and Bevan, 2011*; *Chen et al., 2018*; *McMichael et al., 1983*; *La Gruta and Turner, 2014*; *Kreijtz et al., 2011*), and, to a lesser extent, neutralizing antibodies (*Chen et al., 2018*; *Kreijtz et al., 2011*; *Fang and Sigal, 2005*; *Wang et al., 2015*; *McMichael et al., 1983*) and cytotoxic CD4$^+$ T cells (*Wilkinson et al., 2012*). For the CD8$^+$ T cell response, in particular, it remains unclear whether the efficacy of these cells is dictated by their own density (*Graw and Regoes, 2009*; *Wiedemann et al., 2006*), infected cell density (*Merrill, 1982*; *Caramalho et al., 2009*; *Halle et al., 2016*), or both (*Gadhamsetty et al., 2014*). While quantifying dynamically changing CD8$^+$ T cell efficacy is difficult *in vitro* and *in vivo*, the question is ripe for *in silico* investigation. Several modeling studies have described CD8$^+$ T cell-mediated infected cell clearance for various viral infections in humans and animals, including IAV, HIV, and LCMV (e.g., as in *Ganusov et al., 2011*; *De Boer and Perelson, 1998*; *Perelson and Bell, 1982*; *Miao et al., 2010*; *Cao et al., 2016*; *Lee et al., 2009*; *Price et al., 2015*; *Graw and Regoes, 2009*; *Merrill, 1982*; *Gadhamsetty et al., 2014*; *Baral et al., 2019*; *Bonhoeffer et al., 2000*; *Conway and Perelson, 2015*). Some of these studies have also attempted to link this response and other immune responses to inflammation or disease severity (*Price et al., 2015*; *Baral et al., 2019*; *Manchanda et al., 2014*), but have not yet found the appropriate mathematical relation with the available data. In addition, for IAV infections in particular, varied efficiency of the CD8$^+$ T cell response throughout the course of infection, their early antigen-specific and lung-resident responses (*Yoon et al., 2007*; *McGill and Legge, 2009*; *Keating et al., 2018*), and the balanced consequences on viral loads and host pathology have not yet been investigated in detail.

A better understanding of infected cell clearance may also yield insight into the damage induced to the lung and the ensuing immunopathology during IAV infection. In general, widespread alveolar disease is observed in patients who succumb to the infection (*Bautista et al., 2010*; *Shieh et al., 2010*; *Gao et al., 2013*), and CT scans show bilateral and multi-focal areas of lung damage (e.g. as in *Kohr et al., 2010*; *Li et al., 2011*; *Ajlan et al., 2009*; *Soto-Abraham et al., 2009*; *Gill et al., 2010*; *Li et al., 2012*). Further, hospitalized patients that died as a result from infection with the 2009 H1N1 influenza virus had large numbers of CD8$^+$ T cells present in their lung tissue (*Mauad et al., 2010*). Large pulmonary populations of CD8$^+$ T cells contribute to lung injury by targeting IAV-infected cells (reviewed in *Duan and Thomas, 2016*) and inducing 'bystander damage' to uninfected epithelial cells (*van de Sandt et al., 2017*). However, their population levels do not necessarily indicate active infected cell removal or immunopathology as their extended presence in the lung also contributes to surveillance and repair (*Matheu et al., 2013*; *Sun et al., 2009*). Macrophages and neutrophils also persist in the lung following viral clearance, and their role in inflammation and tissue damage is well documented (reviewed in *Watanabe et al., 2019*; *La Gruta et al., 2007*). A favorable outcome requires a balance between immune-mediated protection and pathology, and the viral-immunological landscape that drives disease and the markers that distinguish mild from severe influenza are complex. This is particularly true in humans (*Tang et al., 2019*) where obtaining high quality data from the lung is challenging, and viral loads in the upper respiratory tract do not always reflect the lower respiratory tract environment (*Feikin et al., 2017*; *Ai et al., 2020*; *Wong et al., 2020*; *Li et al., 2012*; *Ruuskanen et al., 2011*).

The accumulation of damage to the epithelium during IAV infection, either from virally-induced cell lysis or immune-mediated effects, is relatively understudied. We and others have modeled the lung damage and inflammation inflicted during pulmonary infections (e.g., as in *Smith et al., 2011b*; *Reynolds et al., 2006*; *Dunster et al., 2014*; *Ramirez-Zuniga et al., 2019*; *Baral et al., 2019*) but did so without sufficient data to constrain the model formulations. The difficulty in measuring the dynamics of infected cells and in establishing how damage correlates to specific host responses has been the primary impediment. However, recent technological advances, including the use of

reporter viruses (*Manicassamy et al., 2010*; *Tran et al., 2013*; *Karlsson et al., 2015*; *Luker and Luker, 2010*) and lung histomorphometry (*Marathe et al., 2016*; *Marathe et al., 2017*; *Sartorius et al., 2007*; *Zachariadis et al., 2006*; *Boyd et al., 2020*) within animal models, have provided an opportunity to acquire these types of measurements. Whole lung histomorphometry, which is broadly defined as a quantitative microscopic measurement of tissue shape, has recently increased in use due to the ability to directly stain, visualize, and quantify areas of infected cells. Even with these techniques, quantitative data over the course of infection is not currently available. Having data such as these should help reconcile potential nonlinearities in infected cell clearance and provide insight into the accumulated lung inflammation and damage, which are generally thought to be markers of disease severity. In addition, it should bolster the development of robust, predictive computational models, which have historically lacked constraint to these types of data.

In general, disease severity may not be directly correlated to viral loads or specific immunological components. In humans infected with IAV, viral loads typically increase prior to the onset of systemic symptoms, which peak around 2–3 d post-infection (pi) (*Carrat et al., 2008*; *Lee and Lee, 2010*; *Han et al., 2018*). Some symptoms (e.g., fever) are cytokine mediated (*Monto et al., 2000*), but respiratory symptoms often last longer and can remain following viral clearance (*Lee and Lee, 2010*; *Eccles, 2005*). This is particularly true when there is scarring of the lung tissue (*Li et al., 2012*). In the murine model, weight loss is used as an indicator of disease progression and severity, where greater weight loss corresponds to more severe disease (*Trammell and Toth, 2011*; *Bouvier and Lowen, 2010*; *Parzych et al., 2013*). Animal weights generally drop slowly in the first several days during an IAV infection and more rapidly as the infection begins to resolve (*Srivastava et al., 2009*; *Huang et al., 2017*). This is unlike viral load dynamics in these animals, which increase rapidly during the first 0–3 d pi then remain relatively constant prior to resolution (*Smith et al., 2018*). Because weight loss can occur following resolution of the infection, immune-mediated pathology is thought to play a role (*Lauder et al., 2011*; *Xu et al., 2004*; *Ostler et al., 2002*; *Parzych et al., 2013*; *Duan and Thomas, 2016*). Host and pathogen factors, such as age, viral proteins, and inoculum size, have also been shown to influence disease progression (*Toapanta and Ross, 2009*; *Lu et al., 2018*; *Smith et al., 2019*; *McAuley et al., 2007*). While the precise contribution of different factors to IAV-associated disease and mortality remain elusive, having tools that can link these with disease severity and decipher correlation from causation would improve our ability to effectively predict, understand, and treat the disease.

To gain deeper insight into the dynamics of viral resolution during primary pulmonary influenza infection and investigate the connection between viral loads, damage, inflammation, and disease severity, we developed and validated a mathematical model that describes viral kinetics and CD8$^+$ T cell-mediated infected cell clearance. The model accurately predicted the dynamics of effector and memory CD8$^+$ phenotypes, agreed with our previous results that infected cells are cleared in a density-dependent manner with CD8 efficiency rapidly accelerating by over 48,000 times in 2 d, and illuminated tradeoffs between the innate and adaptive immune responses. Our model predicted that infection duration is dependent on the magnitude of CD8$^+$ T cells rather than their efficacy, which we verified by depleting CD8$^+$ T cells. Quantitative whole lung histomorphometry showed that infected areas increase during the first 6 d of infection before clearing rapidly, that the relative number of CD8$^+$ T cells corresponded to the cleared area, and corroborated the model-predicted infected cell dynamics. These data also showed that the infection-induced lung injury is distinct from inflammation, and that each correlates to different immune responses and could be predicted using independent mathematical equations. In addition, the data revealed nonlinear connections between these two pathological measurements and disease severity. Together, our data, model, and analyses provide a robust quantification of the density-dependent nature of CD8$^+$ T cell-mediated clearance, and the critical connections between these cells and the dynamics of viral loads, infected cells, lung injury, inflammation, and disease severity.

## Results

### Virus and CD8$^+$ T cell kinetics

In animals infected with 75 TCID$_{50}$ PR8, virus rapidly infects cells or is cleared within 4 hr pi and is undetectable in all animals at this time (*Figure 1*). Virus then increases exponentially and peaks after

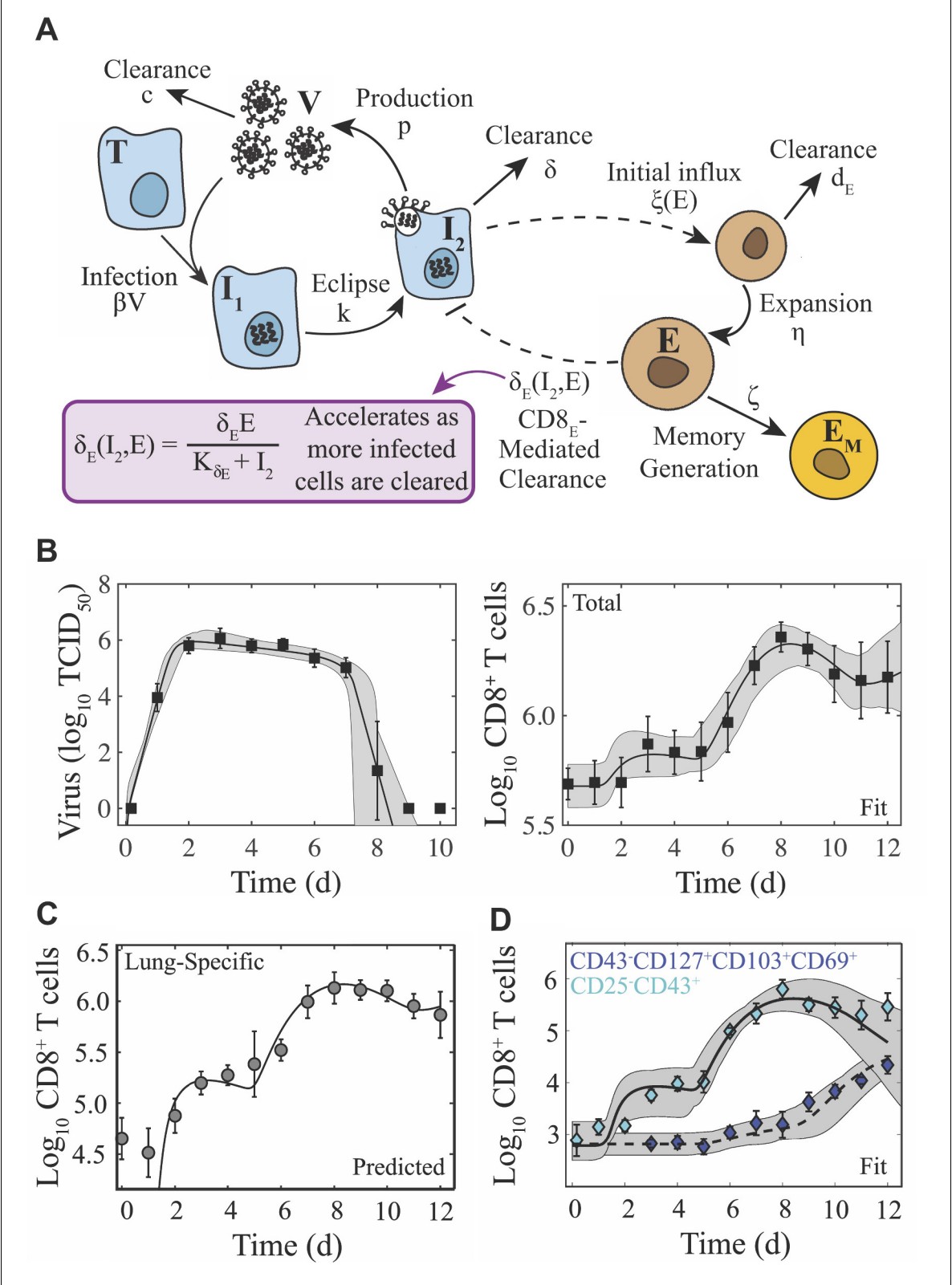

**Figure 1.** Schematic and fit of the CD8[+] T cell viral kinetic model. (**A**) Schematic of the CD8[+] T cell model in *Equation (1)-(6)*. In the model, target cells (*T*) are infected at rate $\beta V$. Infected cells ($I_1$) undergo an eclipse phase and transition to become productively-infected cells ($I_2$) at rate *k*. Virus (*V*) is produced by infected cells at rate *p* and is cleared at rate *c*. Infected cells are cleared at rate $\delta$ by non-specific mechanisms and at rate $\delta_E(I_2, E)$ by effector CD8[+] T cells (*E*; denoted CD8$_E$). The dashed lines represent interactions between infected cells and CD8$_E$. Initial CD8$_E$ influx

*Figure 1 continued on next page*

Figure 1 continued

($\xi(E) = \xi/(K_E + E)$) is proportional to infected cells and is limited by CD8$_E$ with half-saturation constant $K_E$. CD8$_E$ expansion ($\eta$) occurs proportional to infected cells $\tau_E$ days ago. Memory CD8$^+$ T cell ($E_M$; denoted CD8$_M$) generation occurs at rate $\zeta$ and proportional to CD8$_E$ $\tau_M$ days ago. (B) Fit of the CD8$^+$ T cell model (*Equation (1)-(6)*) to virus and total CD8$^+$ T cells from the lungs of mice infected with 75 TCID$_{50}$ PR8 (10 mice per time point). The total number of CD8$^+$ T cells is $\hat{E} = E + E_M + \hat{E}_0$. (C) Total CD8$^+$ T cells in the lung parenchyma (gray circles) and overlay of the model predicted values ($E + E_M$). (D) Fit of the model to virus, CD25$^-$CD43$^+$ CD8$^+$ T cells (cyan diamonds; $E$), and CD43$^-$CD127$^+$CD103$^+$CD69$^+$ CD8$^+$ T cells (blue diamonds; $E_M$) (five mice per time point). The solid and dashed black lines are the optimal solutions and the gray shading is are the model solutions using parameter sets within the 95% CIs. Parameters are given in *Table 1*. Data are shown as mean ± standard deviation.

The online version of this article includes the following figure supplement(s) for figure 1:

**Figure supplement 1.** Dynamics of CD8$^+$ T cells in the lung parenchyma and vasculature.
**Figure supplement 2.** Flow cytometry gating strategy for CD8$^+$ T cell analysis.

---

~2 d pi. Following the peak, virus enters a biphasic decline. In the first phase (2–6 d pi), virus decays slowly and at a relatively constant rate (0.2 log$_{10}$ TCID$_{50}$/d) (*Smith et al., 2018*). In the second phase (7–8 d pi), virus is cleared rapidly with a loss of 4–5 log$_{10}$ TCID$_{50}$ in 1–2 d (average of –3.8 log$_{10}$ TCID$_{50}$/d) (*Smith et al., 2018*). CD8$^+$ T cells remain at their baseline level from 0–2 d pi, where ~15% are located in the circulating blood, ~75% in the lung vasculature, and ~10% in the lung parenchyma (*Figure 1—figure supplement 1*) with the majority in the lung as CD43$^-$CD127$^+$. IAV-specific CD8$^+$ T cells that are recently primed (CD25$^+$CD43$^+$) or effector cells (CD25$^-$CD43$^+$) (*Keating et al., 2018*) begin appearing in the lung and increase slightly from 2– 3 d pi but remain at low levels. These cells are thought to proliferate within the lung at least once by 4 d pi (*McGill and Legge, 2009*), and their populations briefly contract (3–5 d pi) before expanding rapidly (5–8 d pi). During the expansion phase, >95% of recovered cells are in the lung (~70% in the parenchyma and ~30% in the vasculature) with CD25$^+$CD43$^+$ and CD25$^-$CD43$^+$ as the predominant phenotypes. This is in accordance with prior studies that showed these phenotypes are IAV-specific and that their expansion dynamics in the lung were not altered by removing blood-borne CD8$^+$ T cells from the analysis (*Anderson et al., 2012*; *Keating et al., 2018*).

CD8$^+$ T cell expansion corresponds to the second viral decay phase with sixty percent of mice clearing the infection by 8 d pi and the other forty percent by 9 d pi (*Figure 1*). Most CD43$^+$CD8$^+$ T cell phenotypes decline following viral clearance (8–10 d pi) but do not return to their baseline level by 12 d pi. Long-lived antigen-specific memory phenotypes down regulate CD43 (*Jones et al., 1994*; *Harrington et al., 2000*; *Onami et al., 2002*) and gradually increase substantially beginning at 9 d pi with most as CD25$^-$, which is qualitatively similar to other studies (*Harrington et al., 2000*). At 12 d pi, ~55% of CD8$^+$ T cells remain in the lung parenchyma and ~20% in the circulating blood indicating exit from the lung (*Figure 1—figure supplement 1*).

## Viral kinetic model with density-dependent CD8$^+$ T cell-mediated clearance

We previously described the influenza viral load kinetics and biphasic decline during primary pulmonary infection using a density-dependent (DD) model (*Smith et al., 2018*), which assumes that the rate of infected cell clearance increases as the density of infected cells decreases (i.e., $\delta_d(I_2) = \delta_d/(K_\delta + I_2)$). Because the rapid decay of virus is thought to be due to the clearance of infected cells by CD8$^+$ T cells, it is unknown if early CD8$^+$ T cell presence contributes to infected cell clearance, and no model has captured the entire CD8$^+$ T cell time course, we developed a model that describes the dynamics of these cells and their efficiency in resolving the infection (*Equation (1)-(6)*; *Figure 1A*). The model includes equations for effector ($E$, denoted CD8$_E$) and memory ($E_M$, denoted CD8$_M$) CD8$^+$ T cells, and two mechanisms of infected cell clearance. The first mechanism is from unspecified, innate mechanisms, which is relatively constant ($\delta$) and primarily acts during the first viral decay phase (2–6 d pi). The second is the CD8$_E$-mediated infected cell clearance, which occurs at a rate that increases as the density of infected cells decreases ($\delta_E(I_2, E) = \delta_E E/(K_{\delta_E} + I_2)$) and primarily acts during the rapid, second viral decay phase (7–8 d pi). Excluding this density dependence entirely resulted in a significant and premature decline in viral loads, which disagreed with the experimental data. We also tested whether the density dependence could be included in the CD8$^+$ T cell expansion rate rather than in the infected cell clearance rate (see Appendix 1) as other models have done (e.g., as in *Baral et al., 2019*; *Bonhoeffer et al., 2000*; *Conway and*

*Perelson, 2015*). This modification yielded a close fit to the CD8$^+$ T cell data at 6–10 d pi but not at the early time points (*Appendix 1—figure 1*). In addition, the viral load data was underestimated at 7 d pi causing the solution to miss the rapid decline between 7–8 d pi and result in a statistically poorer fit. Thus, retaining the density-dependence in the rate of infected cell clearance most accurately captured the entire dataset. The model includes terms for the initial CD8$_E$ influx at 2–3 d pi ($\xi I_2/(K_E + E)$) and for CD8$_E$ expansion ($\eta E I_2(t - \tau_E)$), which accounts for the larger increase between 5–8 d pi. To capture the contraction of CD8$^+$ T cells between these times (3–5 d pi), it was necessary to assume that the initial CD8$_E$ influx is regulated by their own population (i.e., $\xi(E) = \xi/(K_E + E)$). In both terms, the increase is proportional to the number of infected cells. Although memory CD8$^+$ T cells were not the primary focus here, it was necessary to include the CD8$_M$ population because CD8$^+$ T cells are at a significantly higher level at 10–12 d pi than at 0 d pi (*Figure 1B*). Fitting the model simultaneously to viral loads and total CD8$^+$ T cells from the (non-perfused) lungs of infected animals illustrated the accuracy of the model (*Figure 1B*). The resulting parameter values, 95% confidence intervals (CIs), ensembles, and histograms are given in *Table 1*, *Figure 2*, and *Figure 2—figure supplements 1–2*.

Plotting the model predicted dynamics for the lung-specific CD8$^+$ T cells ($\mathrm{CD8_L} = E + E_M$) illustrated the accuracy of the model in predicting their dynamics within the lung parenchyma without fitting to these data (*Figure 1C*). One benefit of using the total CD8$^+$ T cells is that the model

**Table 1.** CD8$^+$ T cell model parameters.

Parameters and 95% confidence intervals obtained from fitting the CD8$^+$ T cell model (*Equation (1)-(6)*) to viral titers and total CD8$^+$ T cells ('Total CD8') or viral titers, CD25$^-$CD43$^+$CD8$^+$ T cells, and CD43$^-$CD127$^+$CD103$^+$CD69$^+$CD8$^+$ T cells ('Specific CD8$_{E,M}$ Phenotypes') from mice infected with 75 TCID$_{50}$ PR8. CD8$_E$ and CD8$_M$ denote effector ($E$) and memory ($E_M$) CD8$^+$ T cells, respectively. The total number of CD8$^+$ T cells is $\hat{E} = E + E_M + \hat{E}_0$ and is denoted by CD8.

| Parameter | Description | Units | Total CD8 Value | Total CD8 95% CI | Specific CD8$_{E,M}$ phenotypes Value | Specific CD8$_{E,M}$ phenotypes 95% CI |
|---|---|---|---|---|---|---|
| $\beta$ | Virus infectivity | TCID$_{50}^{-1}$ d$^{-1}$ | $6.2 \times 10^{-5}$ | $[5.3 \times 10^{-6}, 1.0 \times 10^{-4}]$ | $3.7 \times 10^{-5}$ | $[1.1 \times 10^{-5}, 9.4 \times 10^{-5}]$ |
| $k$ | Eclipse phase transition | d$^{-1}$ | 4.0 | [4.0, 6.0] | 5.1 | [4.0, 6.0] |
| $p$ | Virus production | TCID$_{50}$ cell$^{-1}$ d$^{-1}$ | 1.0 | $[5.8 \times 10^{-1} \times 1.1 \times 10^{2}]$ | 1.5 | [0.73, 13.6] |
| $c$ | Virus clearance | d$^{-1}$ | 9.4 | $[5.6, 9.5 \times 10^{2}]$ | 12.1 | [5.8, 17.5] |
| $\delta$ | Infected cell clearance | d$^{-1}$ | $2.4 \times 10^{-1}$ | $[1.0 \times 10^{-1}, 6.6 \times 10^{-1}]$ | $3.0 \times 10^{-1}$ | $[1.9 \times 10^{-1}, 5.9 \times 10^{-1}]$ |
| $\delta_E$ | Infected cell clearance by CD8$_E$ | cells CD8$_E^{-1}$ d$^{-1}$ | 1.9 | $[3.3. \times 10^{-1}, 2.0]$ | 5.7 | [1.7, 8.5] |
| $K_{\delta_E}$ | Half-saturation constant | cells | $4.3 \times 10^{2}$ | $[1.0 \times 10^{2}, 2.9 \times 10^{5}]$ | $1.3 \times 10^{2}$ | $[1.0 \times 10^{1}, 8.6 \times 10^{2}]$ |
| $\xi$ | CD8$_E$ infiltration | CD8$_E^2$ cell$^{-1}$ d$^{-1}$ | $2.6 \times 10^{4}$ | $[1.3 \times 10^{2}, 8.7 \times 10^{4}]$ | $2.9 \times 10^{3}$ | $[8.0 \times 10^{2}, 1.7 \times 10^{4}]$ |
| $K_E$ | Half-saturation constant | CD8$_E$ | $8.1 \times 10^{5}$ | $[1.0 \times 10^{3}, 1.0 \times 10^{6}]$ | $2.2 \times 10^{6}$ | $[1.1 \times 10^{6}, 8.3 \times 10^{6}]$ |
| $\eta$ | CD8$_E$ expansion | cell$^{-1}$ d$^{-1}$ | $2.5 \times 10^{-7}$ | $[1.6 \times 10^{-8}, 6.7 \times 10^{-7}]$ | $3.5 \times 10^{-7}$ | $[2.3 \times 10^{-7}, 5.2 \times 10^{-7}]$ |
| $\tau_E$ | Delay in CD8$_E$ expansion | d | 3.6 | [2.1, 5.9] | 3.3 | [2.6, 3.8] |
| $d_E$ | CD8$_E$ clearance | d$^{-1}$ | 1.0 | $[5.1 \times 10^{-2}, 2.0]$ | 1.1 | $[3.3 \times 10^{-1}, 2.5]$ |
| $\zeta$ | CD8$_M$ generation | CD8$_M$ CD8$_E^{-1}$ d$^{-1}$ | $2.2 \times 10^{-1}$ | $[1.0 \times 10^{-2}, 9.4 \times 10^{-1}]$ | $3.2 \times 10^{-2}$ | $[1.0 \times 10^{-2}, 2.2 \times 10^{-1}]$ |
| $\tau_M$ | Delay in CD8$_M$ generation | d | 3.5 | [3.0, 4.0] | 3.3 | [2.1, 5.3] |
| $\hat{E}_0$ | Baseline CD8 or CD8$_E$ | CD8 or CD8$_E$ | $4.2 \times 10^{5}$ | $[3.3. \times 10^{5}, 5.3 \times 10^{5}]$ | $6.0 \times 10^{2}$ | $[3.1 \times 10^{2}, 1.8 \times 10^{2}]$ |
| $\hat{E}_{M0}$ | Baseline CD8$_M$ | CD8$_M$ | - | - | $6.5 \times 10^{2}$ | $[3.2 \times 10^{2}, 1.5 \times 10^{3}]$ |
| $T(0)$ | Initial uninfected cells | cells | $1 \times 10^{7}$ | - | $1 \times 10^{7}$ | - |
| $I_1(0)$ | Initial infected cells | cells | 75 | - | 75 | - |
| $I_2(0)$ | Initial infected cells | cells | 0 | - | 0 | - |
| $V(0)$ | Initial virus | TCID$_{50}$ | 0 | - | 0 | - |
| $E(0)$ | Initial CD8$_E$ | CD8$_E$ | 0 | - | 0 | - |
| $E_M(0)$ | Initial CD8$_M$ | CD8$_M$ | 0 | - | 0 | - |

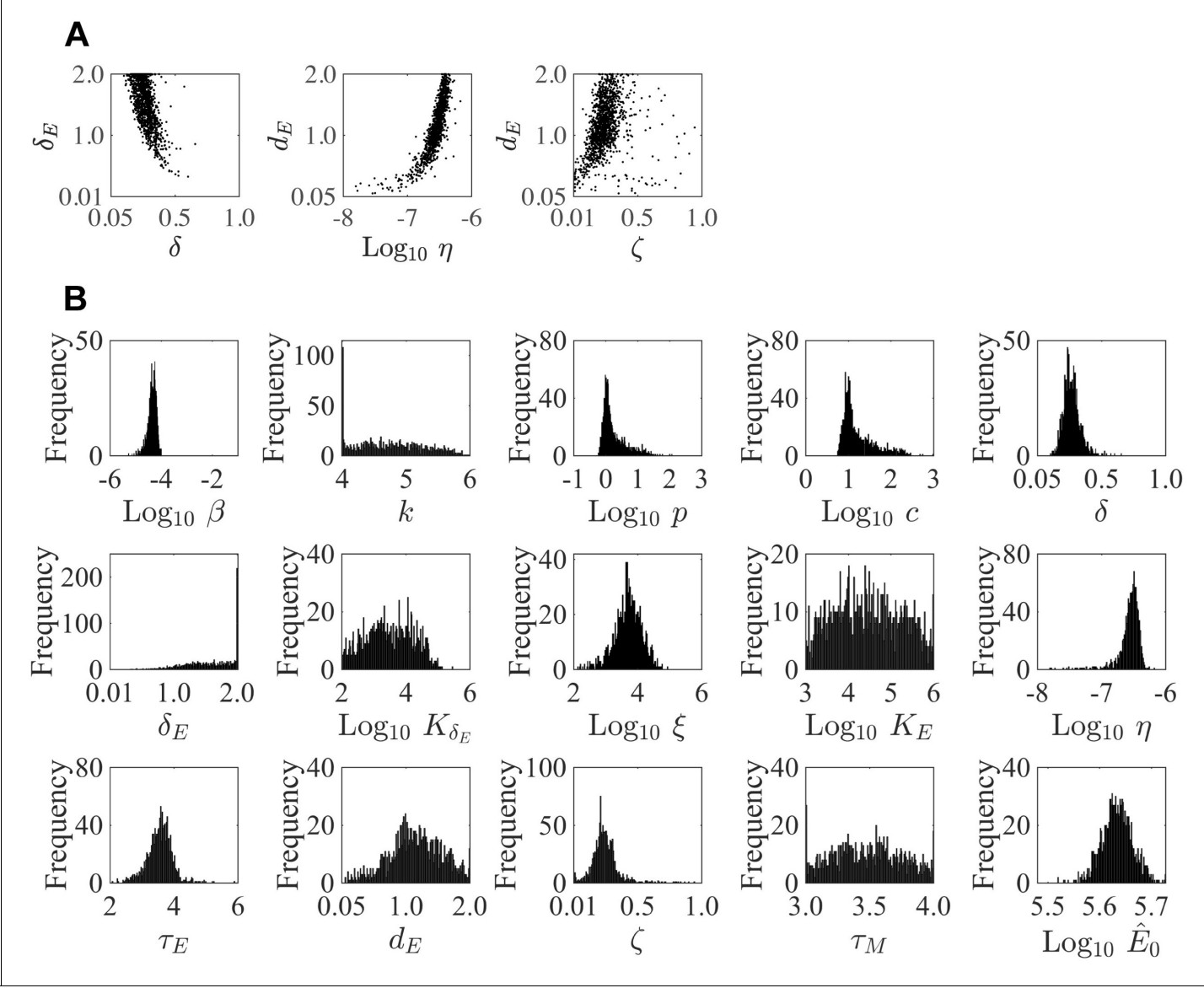

**Figure 2.** Parameter ensembles and histograms. Parameter ensembles (A) and histograms (B) resulting from fitting the CD8$^+$ T cell viral kinetic model (*Equation (1)-(6)*) to viral titers and total CD8$^+$ T cells from mice infected with 75 TCID$_{50}$ PR8. (A) The rates of infected cell clearance by non-specific mechanisms ($\delta$) and by CD8$_E$ ($\delta_E$) are slightly negatively correlated. Correlations were also present between the rates of CD8$_E$ clearance ($d_E$), CD8$_E$ expansion ($\eta$), and CD8$_M$ generation ($\zeta$). The axes limits reflect imposed bounds. Additional ensemble plots are in *Figure 2—figure supplements 1–2*. (B) The histograms show that the majority of parameters are well-defined with the exception of the eclipse phase transition rate ($k$), one of the half-saturation constants ($K_E$), and the CD8$_M$ generation delay ($\tau_M$).

The online version of this article includes the following figure supplement(s) for figure 2:

**Figure supplement 1.** Parameter ensembles.

**Figure supplement 2.** Parameter ensembles.

automatically deduces the dynamics of effector-mediated killing and memory generation without needing to specify which phenotypes might be involved in these processes as they may be dynamically changing. However, the rates of expansion, contraction, and infected cell clearance may be different if only certain phenotypes are engaged. Thus, we examined whether the model could fit the dynamics of the predominant effector (CD25$^-$CD43$^+$) and memory (CD43$^-$CD127$^+$CD103$^+$CD69$^+$) phenotypes. Re-fitting the model to these data suggested that no changes to model formulation

were needed and there were only small alterations to select CD8-specific parameter values (**Figure 1D**; **Table 1**).

Plotting the model ensembles revealed a correlation between the two infected cell clearance parameters ($\delta$ and $\delta_E$; **Figure 2B**), which represent the efficacy of the non-specific immune response and the CD8$^+$ T cell response, respectively. Performing a sensitivity analysis showed that the viral load dynamics do not change substantially when these parameters are increased or decreased (**Appendix 2—figure 1**) . However, decreasing the rate of non-specific infected cell clearance (i.e., lower $\delta$) resulted in a significant increase in the number of CD8$_E$ due to the small increase in the number of infected cells (**Figure 1**). Even with a larger CD8$_E$ population, recovery was delayed by only ~0.1 d. Given the correlation between $\delta$ and $\delta_E$ (**Figure 2B**), a more efficient CD8$_E$ response (i.e., higher $\delta_E$) may be able to overcome this short delay in resolution. The lack of sensitivity to changes in the infected cell clearance parameters is in contrast to the DD model, where the viral dynamics were most sensitive to perturbations in $\delta_d$ (**Appendix 2—figure 1**; **Smith et al., 2018**), which encompasses multiple processes. With CD8$^+$ T cells explicitly included in the model, the infection duration was most sensitive to changes in the rate of CD8$_E$ expansion ($\eta$) (**Figure 1**, **Appendix 1—figure 1**; discussed in more detail below).

Examining the parameter ensembles and sensitivity analysis also yielded insight into how other model parameters affect the CD8$^+$ T cell response. The rates of CD8$_E$ expansion ($\eta$) and clearance ($d_E$) were slightly correlated, indicating a balance between these two processes (**Figure 2A**). This correlation and the sensitivity of $\eta$ produced model dynamics that were also sensitive to changes in the CD8$_E$ clearance rate ($d_E$) (**Appendix 3—figure 2**). As expected, the rates of CD8$_M$ generation ($\zeta$) and CD8$_E$ clearance ($d_E$) were correlated (**Figure 2A**). It has been estimated that approximately 5–10% of effector CD8$^+$ T cells survive to become a long-lasting memory population (**Kaech et al., 2002**). Despite the inability to distinguish between CD8$_E$ and CD8$_M$ in the total CD8$^+$ T cell data, the model predicts that 17% of CD8$_E$ transitioned to a memory class by 15 d pi. When considering only the CD25$^-$CD43$^{+/-}$ effector and memory phenotypes, the model estimates this value to be ~7%. Additional insight into the regulation of the CD8$^+$ T cell response, results from the model fitting, and a comparison of the DD model and the CD8$^+$ T cell model are included in Appendix 2.

## Density-dependent infected cell clearance

Given the accuracy of the model, we next sought to gain further insight into the nonlinear dynamics of CD8$^+$ T cell-mediated infected cell clearance. Plotting the clearance rate ($\delta_E(I_2, E) = \delta_E E/(K_{\delta_E} + I_2)$) as a function of infected cells ($I_2$) and CD8$_E$ ($E$) (**Figure 3**) confirmed that there is minimal contribution from CD8$_E$-mediated clearance to viral load kinetics or infected cell kinetics prior to 7 d pi (**Figure 3A–C**, markers a–b). At the initiation of the second decay phase (7 d pi), the clearance rate is $\delta_E(I_2, E) = 3.5/\text{d}$ (**Figure 3A–B**, marker c). As the infected cell density declines toward the half-saturation constant ($K_{\delta_E} = 4.3 \times 10^2 \text{ cells}$), the clearance rate increases rapidly to a maximum of 4830/d (**Figure 3A–C**, markers d–g). The model predicts that there are $6 \times 10^5$ infected cells remaining at 7 d pi, which can be eliminated by CD8$_E$ in 6.7 h.

To explore how recovery time is altered by varying CD8$_E$ levels, we examined the resulting dynamics from increasing or decreasing the rate of CD8$_E$ expansion ($\eta$). When $\eta$ was increased by 50%, the CD8$_E$ population increased by a substantial 670% (**Appendix 3—figure 1**). However, this was insufficient to significantly shorten the infection (8.4 d versus 7.8 d). The infection duration could be reduced if CD8$_E$ expansion began earlier (i.e., smaller $\tau_E$; **Appendix 3—figure 2**). Although recovery is not significantly expedited by a larger CD8$_E$ population, our model predicted that the infection would be dramatically prolonged if these cells are sufficiently depleted (**Figure 3D–E** and **Figure 1**). This *in silico* analysis revealed a bifurcation in recovery time, such that the infection is either resolved within ~15 d pi or may last up to ~45 d if CD8$_E$ are below a critical magnitude required to resolve the infection (**Figure 3D–E**). The critical number of total CD8$^+$ T cells needed for successful viral clearance was $\hat{E}_{\max}^{\text{crit}} = 7.4 \times 10^5$ CD8, which was 39.2% of the maximum number of CD8$^+$ T cells obtained from the best-fit solution (i.e., $\hat{E}_{\max} = 1.9 \times 10^6$ CD8). This corresponds to 17% of CD8$_E$ (i.e., $E_{\max}^{\text{crit}} = 2.3 \times 10^5$ CD8$_E$; **Figure 3D–E**). The model analysis indicated that decreasing the total number of CD8$^+$ T cells by as little as 0.1% from this critical level (i.e., 39.2% to 39.1%) lengthened the infection from 15 d pi to 25 d pi (**Figure 3D**).

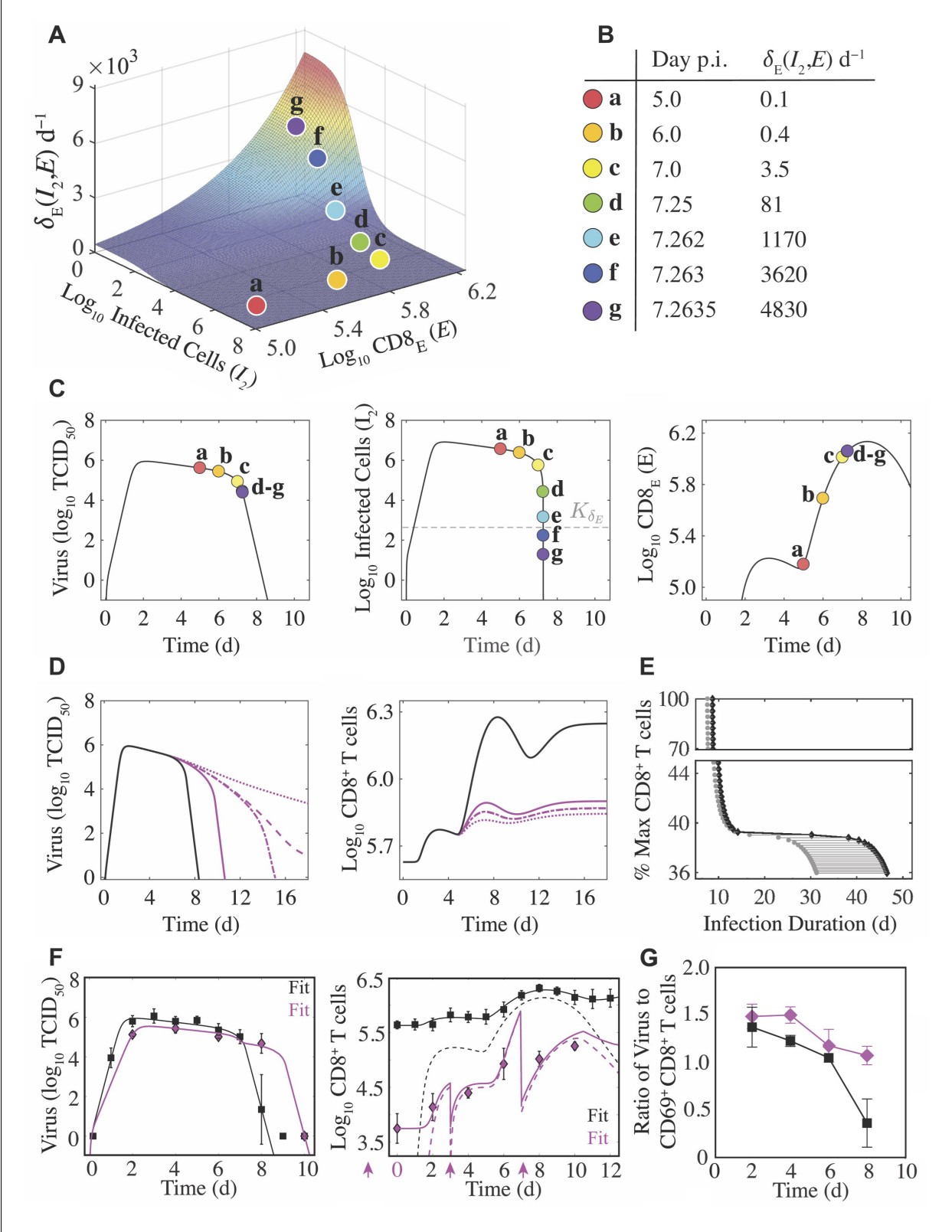

**Figure 3.** Density-dependent infected cell clearance by CD8+ T cells and their impact on recovery time. (A) The rate of CD8$_E$-mediated infected cell clearance ($\delta_E(I_2, E) = \delta_E E/(K_{\delta_E} + I_2)$) plotted as a function of infected cells ($I_2$) and effector CD8+ T cells ($E$; CD8$_E$). The colored markers (denoted a–g) indicate the infected cell clearance rate that corresponds to different time points during the infection for the best-fit solution. (B) Values of $\delta_E(I_2, E)$ for the indicated time points associated with the markers a–g. (C) Corresponding locations of the various $\delta_E(I_2, E)$ values (markers a–g) on the best-fit

*Figure 3 continued on next page*

Figure 3 continued

solution of the CD8$^+$ T cell model for virus ($V$), infected cells ($I_2$), and CD8$_E$ ($E$). (D) Solutions of the CD8$^+$ T cell model (*Equation (1)-(6)*) for virus ($V$) and total CD8$^+$ T cells ($\hat{E}$) using the best-fit parameters (black line) and when varying the CD8$_E$ expansion rate ($\eta$; magenta lines) to illustrate how different total CD8$^+$ T cell magnitudes alter infection duration. The magenta lines are solutions from when the percent $\hat{E}_{\max}$ relative to $\hat{E}_{\max}$ from the best-fit solution was 42% (solid line), 39.2% (dash-dotted line), 39.1% (dashed line), or 37% (dotted line). (E) The time at which infected cells reach the half-saturation constant ($I_2 = K_{\delta_E}$; gray circles) and the infection duration (time where $\log_{10} V = 0$; black diamonds) are shown for the various CD8$^+$ T cell magnitudes. The gray line between these points is the time required to eliminate $K_{\delta_E}$ infected cells and achieve complete resolution of the infection ($\log_{10} V = 0$). (F) Fit of the CD8$^+$ T cell model (*Equation (1)-(6)*) to viral loads and CD8$^+$ T cells (magenta diamonds) following depletion at $-2$, 0, 3, and 7 d pi (magenta arrows). The best model (*Supplementary file 2*) resulted in fewer target cells ($T_0$), a lower CD8$_E$ influx ($\xi$), and a higher CD8$_E$ expansion rate ($\eta$). All other parameters were fixed to the best-fit value in *Table 1*. The solid lines are $\hat{E} = E + E_M + \hat{E}_0$ and the dashed lines are $E$ for the cases where CD8$^+$ T cells were depleted (magenta) and where they were not depleted (black). (G) Comparison of the $\log_{10}$ ratio of virus to CD69$^+$CD8$^+$ T cells with and without CD8$^+$ T cell depletion (magenta and black, respectively). All data are shown as mean ± standard deviation.

## Dynamical changes during CD8$^+$ T cell depletion

To further test our model formulation and identify how viral load kinetics are altered with dynamically changing CD8$^+$ T cells, we infected groups of mice with 75 TCID$_{50}$ and depleted these cells at $-2$ d, 0 d, 3 d, and 7 d pi with an anti-CD8$\alpha$ antibody (clone 2.43; *Figure 3F*). CD8$^+$ T cells were reduced with over 99% efficiency and only 1.3% remained 2 d after depletion in the absence of infection (i.e., at 0 d pi; *Figure 3F*). CD8$^+$ T cells remained >1 $\log_{10}$ lower throughout the infection. Unexpectedly, the corresponding viral loads were significantly lower at 2 d pi (p=0.02) and consistently lower at other time points early in the infection (4 d pi (p=0.1) and 6 d pi (p=0.2)). As expected and predicted by our mathematical model, viral loads were significantly higher at 8 d pi (4.68 $\log_{10}$ TCID$_{50}$ compared to 1.42 $\log_{10}$ TCID$_{50}$; p=0.01). By 10 d pi, a sufficient number of CD8$^+$ T cells were present and all animals had cleared the infection (*Figure 3F*). Interestingly, the number of CD8$^+$ T cells at 10 d pi was only slightly higher than at 8 d pi (5.25 $\log_{10}$ versus 5.02 $\log_{10}$; p=0.064) further highlighting the density-dependent dynamics described above.

Given that viral loads were lower at early time points and that the anti-CD8$\alpha$ antibody is known to cause concentration-dependent changes in CD8$_E$ differentiation, activation, and efficacy (*Cross et al., 2019*) in addition to resulting in death of the cells that would initiate activation of and removal by other immune cells, we did not expect our model to match the data without modulation of parameter values. However, we did expect that no changes to the model formulation would be required. In total, we tested >30 'models' where 1–4 parameters were altered and found one model that was significantly better according to the AIC (*Supplementary file 1*). In that model, the initial number of target cells ($T_0$) was 2.5x lower (~$4 \times 10^6$ versus $1 \times 10^7$ cells), the rate of initial CD8$_E$ influx ($\xi$) was 2x lower (1.3 $\times$ 10$^4$ versus $2.6 \times 10^4$ CD8$_E^2$ cell$^{-1}$ d$^{-1}$), and the rate of CD8$_E$ expansion ($\eta$) was 4x higher ($1 \times 10^{-6}$ versus $2.5 \times 10^{-7}$ cell$^{-1}$ d$^{-1}$). The second best model had a lower cost but was penalized by an additional parameter. That model suggested similar results but replaced the effect on $T_0$ with a combination of a lower virus production rate ($p$) and higher non-specific infected cell clearance rate ($\delta$). Both models resulted in fewer infected cells and, thus, have approximately equivalent interpretations.

To further examine these findings, we plotted the ratio of virus to activated (CD69$^+$) CD8$^+$ T cells (*Figure 3G*). At 2 d pi, the ratio was similar for the depleted and mock-treated groups (p=0.41) suggesting that the depletion-induced reduction in virus was proportional to the reduction in activated CD8$^+$ T cells. However, at 4 d pi, the ratio in the depleted groups was significantly higher (p=0.0028) for otherwise similar levels of virus. This signifies that there were disproportionately low numbers of CD8$^+$ T cells and, thus, reduced influx (i.e., lower $\xi$). By 6 d pi, the ratios were again similar (p=0.25) implying a higher expansion of these cells (i.e., higher $\eta$). Investigation into the predicted target cell reduction is included below.

## Modeling lung injury dynamics

To investigate the dynamics of infected cells and their clearance by CD8$^+$ T cells, we quantified these cells and the progression and resolution of lung injury using serial whole lung histomorphometry (*Figure 4A*). Antigen-positive areas of the lung ('active' lesions) were first detectable at 2 d pi (*Figure 4A–B*), which coincides with the peak in viral loads. The infected area continued to increase

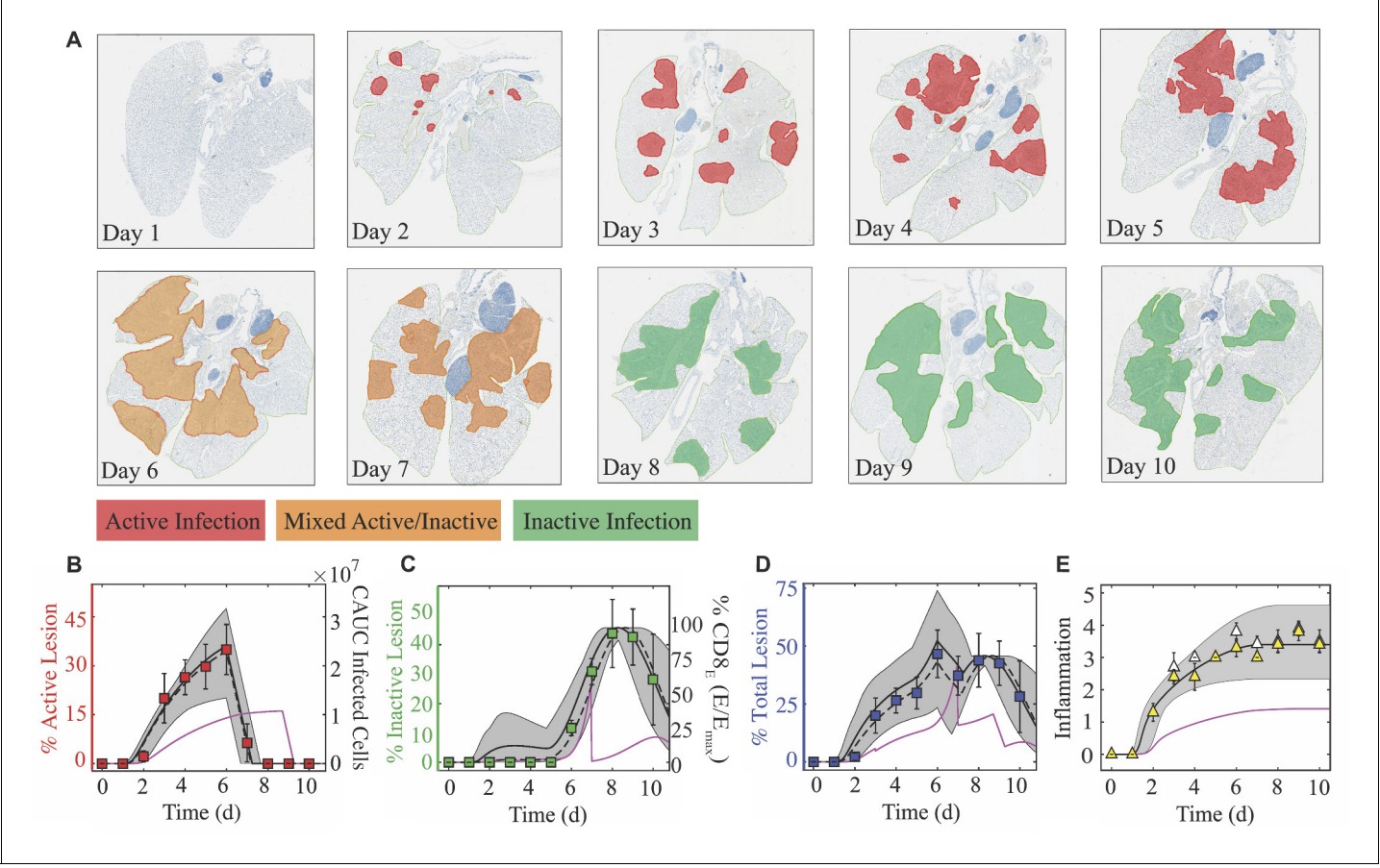

**Figure 4.** Histomorphometry and inflammation of the IAV-infected lung. (A) Whole lung sections with histomorphometry showing the areas of influenza NP-positive 'active' lesions (red), previously infected 'inactive' lesions with minimal antigen-positive debris (green), or mixed active and inactive regions (orange) throughout the infection. Representative images from each group are shown. (B) Percent active lesion (red squares) plotted together with the cumulative area under the curve (CAUC) of the predicted infected cell dynamics ($I_2$) obtained from fitting the CD8$^+$ T cell model. The linear decline in the active lesion (−28.7%/d; see *Figure 4—figure supplement 1*) was used to estimate the decline after 6 d pi. (C) Percent inactive lesion (green squares) plotted together with the percent maximum CD8$_E$ ($E/E_{max}$) obtained from fitting the CD8$^+$ T cell model. (D) The total lesion (blue squares) is the addition of the active and inactive lesions. To include all measurements on the same scale, the CAUC of $I_2$ was multiplied by a scaling factor of 14.2% per $1 \times 10^7$ cells, and the percent maximum CD8$_E$ was multiplied by a scaling factor of 0.46%. (E) Fit of *Equation (7)* to the alveolar (white triangles) and interstitial (yellow triangles) inflammation scores. The solid black, dashed black, and solid magenta lines are the curves generated using the best-fit parameters obtained from fitting the model to the total CD8$^+$ T cells, the CD25$^-$CD43$^+$ and CD43$^-$CD127$^+$CD103$^+$CD69$^+$ CD8$^+$ T cells, and the total CD8$^+$ T cells during CD8 depletion, respectively. The gray shading are the curves generated using the 95% CI parameters from fitting the model to the total CD8$^+$ T cells. All data are shown as mean ± standard deviation.

The online version of this article includes the following figure supplement(s) for figure 4:

**Figure supplement 1.** Regression analysis of lung injury dynamics and CD8$^+$ T cells.

**Figure supplement 2.** Correlation of inflammation with macrophages and neutrophils.

in a nonlinear manner until 6 d pi, whereas viral loads remained high until 7 d pi (*Figure 1B*). At this time, resolution of the infection began and the infected area declined at a rate of ~28.7%/d between 6–7 d pi (*Figure 4—figure supplement 1*). Few to no infected cells were present at 8 d pi (*Figure 4A*). Correspondingly, virus was undetectable in most animals by 8 d pi (*Figure 1B*). Because the percent active lesion is a reflection of the influenza-positive cells, we examined whether the CD8$^+$ T cell model accurately predicted these dynamics. In the model, the accumulated infection is defined by the cumulative area under the curve (CAUC) of the productively infected cells ($I_2$). Plotting the percent active lesion against the CAUC of $I_2$ shows that the model accurately reflects the cumulative infected cell dynamics and, thus, the infection progression within the lung (*Figure 4B*). Plotting the CAUC of $I_2$ for all parameter sets in the 95% CIs and from fitting the model to specific

phenotypes further illustrates the accuracy by showing that the heterogeneity in the histomorphometry data is captured (*Figure 4B*), which is larger than the heterogeneity in viral loads (*Figure 1B*). Plotting the predicted active lesion dynamics for the model parameterized to the data where CD8$^+$ T cells were depleted suggested that there was a ~22% reduction in the active lesion (*Figure 4B*).

Antigen-negative, previously-infected or damaged areas of the lung ('inactive' lesions) are evident beginning at 5 d pi (*Figure 4A,C*). This resolution of the infection continued from 5-8 d pi, causing a 15.1%/d increase in the area of inactive lesions (*Figure 4—figure supplement 1*). Following this, healing of the injured lung is apparent as the inactive lesioned area declines (−14.5%/d from 9–10 d pi; *Figure 4C* and *Figure 4—figure supplement 1*). These dynamics generally parallel the CD8$^+$ T cell dynamics but are nonlinearly correlated (*Figure 4—figure supplement 1*). Fitting a line to the CD8$^+$ T cell data from 5-8 d pi indicated that the influx rate of all phenotypes is $4.94 \times 10^5$ cells/d, of lung-specific phenotypes is $3.97 \times 10^5$ cells/d, and of the CD25$^-$CD43$^+$ effector phenotype is $1.98 \times 10^5$ cells/d (*Figure 4—figure supplement 1*). Thus, the model estimates that, on average, every 100,000 total, lung-specific, or CD25$^-$CD43$^+$ CD8$^+$ T cells clear ~3.1%, ~3.8%, or ~7.6% of the infected areas within the lung, respectively. During the CD8$^+$ T cell contraction phase, a similar linear regression analysis suggested that these cells decline at rates of $\sim 4.13 \times 10^5$ CD8/d (total), $\sim 2.35 \times 10^4$ CD8/d (lung-specific), and $\sim 3.83 \times 10^4$ CD8/d (CD25$^-$CD43$^+$) (*Figure 4—figure supplement 1*). Similar to the relation discussed above, the dynamics of the damaged areas of the lung corresponded to the dynamics of the percent maximum CD8$_E$ (i.e., $E/E_{max}$) in the model (*Figure 4C*). Our model suggested a ~37% reduction in the inactive lesion during CD8$^+$ T cell depletion (*Figure 4C*). Adding the predicted dynamics for the active and inactive lesions agrees with the dynamics of the total lesion, but is slightly underestimated when using the model fit to CD25$^-$CD43$^+$ CD8$^+$ T cells (*Figure 4D*). In addition, the predicted total lesion was reduced for the case where CD8$^+$ T cells were depleted (*Figure 4D*).

## Modeling lung inflammation dynamics

In addition to measuring virus-induced lung damage, lung inflammation was scored (*Figure 4E*). Both alveolar and interstitial inflammation begin to increase at 2 d pi with the sharpest increase between 1–3 d pi. Inflammation continues to increase until 6 d pi with a maximum score of 3.5–4.0 out of 5. Resolution of inflammation was not evident during the time course of our data, which concluded at 10 d pi. This is in contrast to the lung damage inflicted by the virus, which begins declining at 8 d pi and shows that ~15% of the damage was repaired by 10 d pi. Inflammation was linearly correlated to inflammatory macrophages and to neutrophils (*Figure 4—figure supplement 2*), which were excluded from the model. However, the knowledge that the model's predicted infected cell dynamics are accurate and that these cells can produce cytokines and chemokines that attract cells like macrophages and neutrophils suggested that our model could be used to estimate inflammation. To do this, we fit *Equation (7)* to the inflammation scores while keeping all other parameters fixed to their best-fit values (*Table 1*). The results suggested that the equation captured the inflammation dynamics, and that the contribution from the initial infected cell class ($I_1$) was $\alpha_1 = 4.27 \text{ per } 10^7 \text{ cells/d}$ and the contribution from the productively infected cells ($I_2$) was $\alpha_2 = 0.87 \text{ per } 10^7 \text{ cells/d}$. For the case where CD8$^+$ T cells were depleted, *Equation (7)* estimated the inflammation scores to be reduced to ~1.5 (*Figure 4E*).

## Weight loss relates nonlinearly to lung injury and inflammation

To monitor disease progression, weight loss was measured daily throughout the course of infection (*Figure 5*). During the first 5 d pi, animals gradually lost ~4% of their initial weight. This was followed by a sharper drop (8%) at 6 d pi. Animal weights increased slightly at 7 d pi (~6%) before reaching peak weight loss (10–14%) at 8 d pi. Following virus resolution, the animals' weights began to restore as the inactive lesions resolved (9–10 d pi; *Figure 5A*). Weight loss was reduced during CD8$^+$ T cell depletion (*Figure 5A*). Plotting weight loss against the percent total (active and inactive) lesioned area of the lung shows the similarity in their dynamics (*Figure 5A*) and revealed a nonlinear relation (*Figure 5B*). To further quantify their relationship, we fit the saturating function $L(W) = l_{max}W^n/(W^n + K_w^n)$ to these data, where $L$ is the percent total lesioned area, $W$ is percent weight loss, $l_{max}$ is the maximum rate of the interaction, $K_w$ is the half-saturation constant, and $n$ is the Hill coefficient. This function provided a close fit to the data ($R^2 = 0.92$; black line in *Figure 5B*)

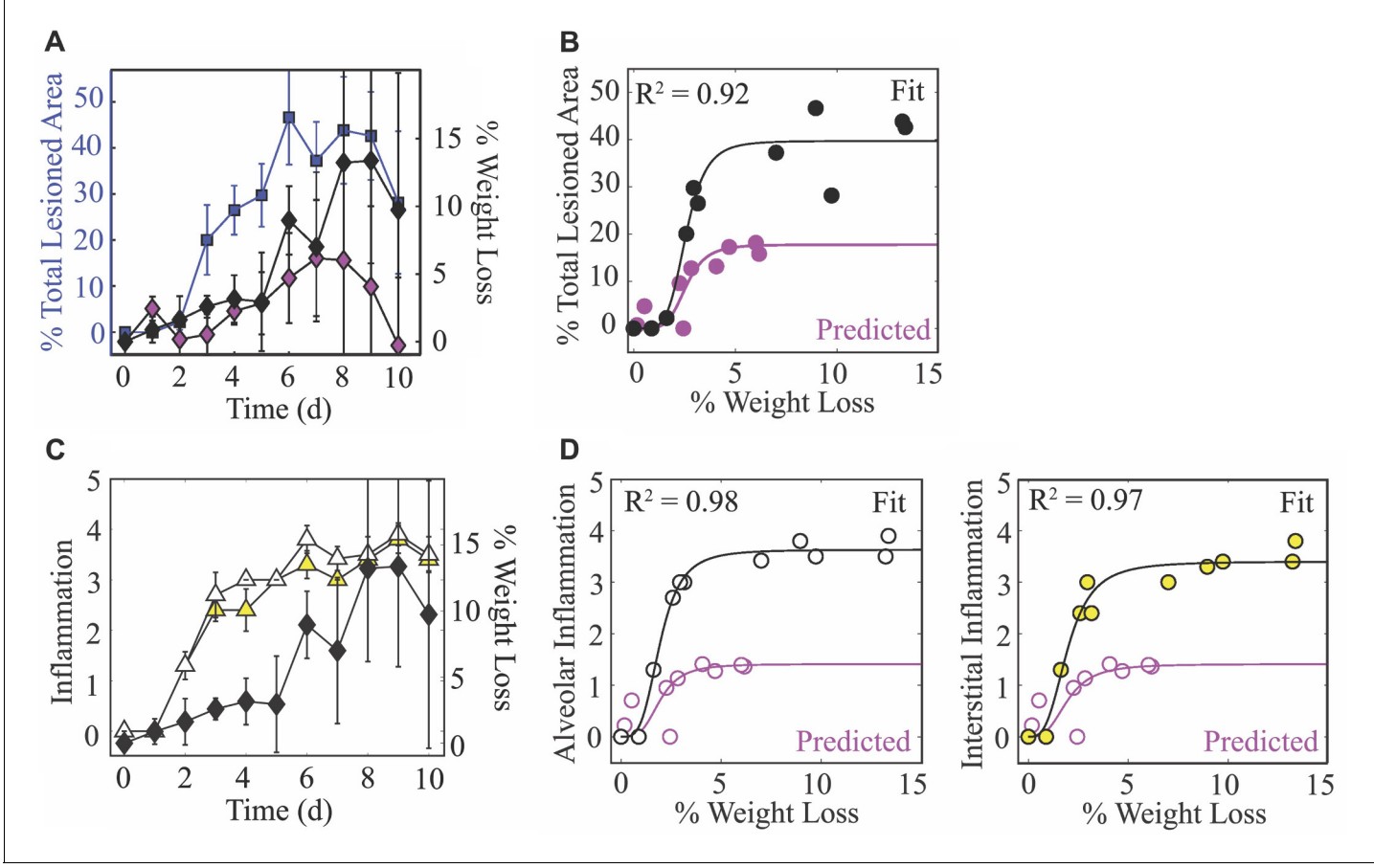

**Figure 5.** Weight loss dynamics and its relation to lung injury and inflammation. (**A**) The percent total (active and inactive) lesion (blue squares) plotted together with the percent weight loss (black diamonds) to illustrate their similar dynamics. (**B**) Fit of a saturating function ($L(W) = l_{max} W^n / (K_w^n + W^n)$; black line) to the mean percent total lesioned area ($L$) and mean weight loss ($W$) for all time points (black circles). The best-fit parameters were $l_{max} = 39.7\%$ total lesioned area, $K_w = 2.6\%$ weight loss, and $n = 5.2$. The magenta circles are the predicted percent total lesioned area (**Figure 4D**) with the corresponding weight loss during CD8 depletion, and the magenta line is the prediction using the best-fit parameters ($K_w = 2.6\%$ weight loss and $n = 5.2$) together with the maximum predicted percent total lesion ($l_{max} = 17.73\%$ total lesioned area). (**C**) The alveolar (white triangles) and interstitial (yellow triangles) inflammation plotted together with the percent weight loss (black diamonds). (**D**) Fit of a saturating function (black line) to the mean alveolar (white circles) and interstitial (yellow circles) inflammation scores and mean weight loss for all time points. The best-fit parameters for alveolar inflammation (white circles; black line) were $l_{a_{max}} = 3.63$ score, $K_{a_w} = 1.95\%$ weight loss, and $n_a = 3.65$, and for interstitial inflammation were $l_{i_{max}} = 3.40$ score, $K_{i_w} = 1.96\%$ weight loss, and $n_i = 3.15$. The magenta circles are the predicted percent inflammation score (**Figure 4E**) with the corresponding weight loss during CD8 depletion, and the magenta line is the prediction using the best-fit parameters ($K_{a_w} = 1.95\%$ weight loss, $n_a = 3.65$; $K_{i_w} = 1.96\%$ weight loss, $n_i = 3.15$) together with the maximum predicted inflammation ($l_{max} = 1.41$ score).

with best-fit parameters $l_{max} = 39.7\%$ total lesioned area, $K_w = 2.58\%$ weight loss, and $n = 5.24$. Plotting the estimated percent total lesion during CD8[+] T cell depletion (**Figure 4D**) together with the measured weight loss (**Figure 5A**) also supported the nonlinear relation. Using the same best-fit parameters ($K_w = 2.58\%$ weight loss and $n = 5.24$) together with the predicted maximum percent total lesion ($l_{max} = 17.73\%$ total lesioned area) showed the accuracy of this function (magenta line in **Figure 5D**). Weight loss was also nonlinearly correlated to the alveolar and interstitial inflammation scores (**Figure 5C**), although the relation was slightly different compared to that for the lung injury data. Fitting the same function to these data independently for alveolar and interstitial inflammation also provided a close fit ($R^2 = 0.98$ and $R^2 = 0.97$; black lines in **Figure 5D**). Best-fit parameters for alveolar inflammation were $l_{a_{max}} = 3.63$ score, $K_{a_w} = 1.95\%$ weight loss, and $n_a = 3.65$. Best-fit parameters for interstitial inflammation were $l_{i_{max}} = 3.40$ score, $K_{i_w} = 1.96\%$ weight loss, and $n_i = 3.15$. For the CD8-depleted case, plotting the estimated inflammation (**Figure 4E**) together with the measured weight loss (**Figure 5A**) again supported the nonlinear relation. In addition, using the same

best-fit parameters ($K_{a_w} = 1.95\%$ weight loss, $n_a = 3.65$; $K_{i_w} = 1.96\%$ weight loss, $n_i = 3.15$) together with the predicted maximum inflammation score ($l_{i_{max}} = 1.41$ score) showed the accuracy (magenta line in *Figure 5D*).

## Discussion

Influenza A virus infections pose a significant threat to human health, and it is crucial to understand and have tools that can predict how the virus spreads within the lower respiratory tract, how specific immune responses contribute to infection control, and how these relate to disease progression. Although it has been difficult to directly relate these features and obtain high quality data from the lower respiratory tract in humans, we circumvented the challenge by pairing comprehensive experimental data with robust mathematical models and analyses. Our iterative model-driven experimental approach (*Smith, 2018a*; *Smith, 2018b*) revealed important dynamic relations between virus, infected cells, immune cells, lung damage, inflammation, and disease severity (summarized in *Figure 6*). Identifying these nonlinear connections allows for more accurate interpretations of viral

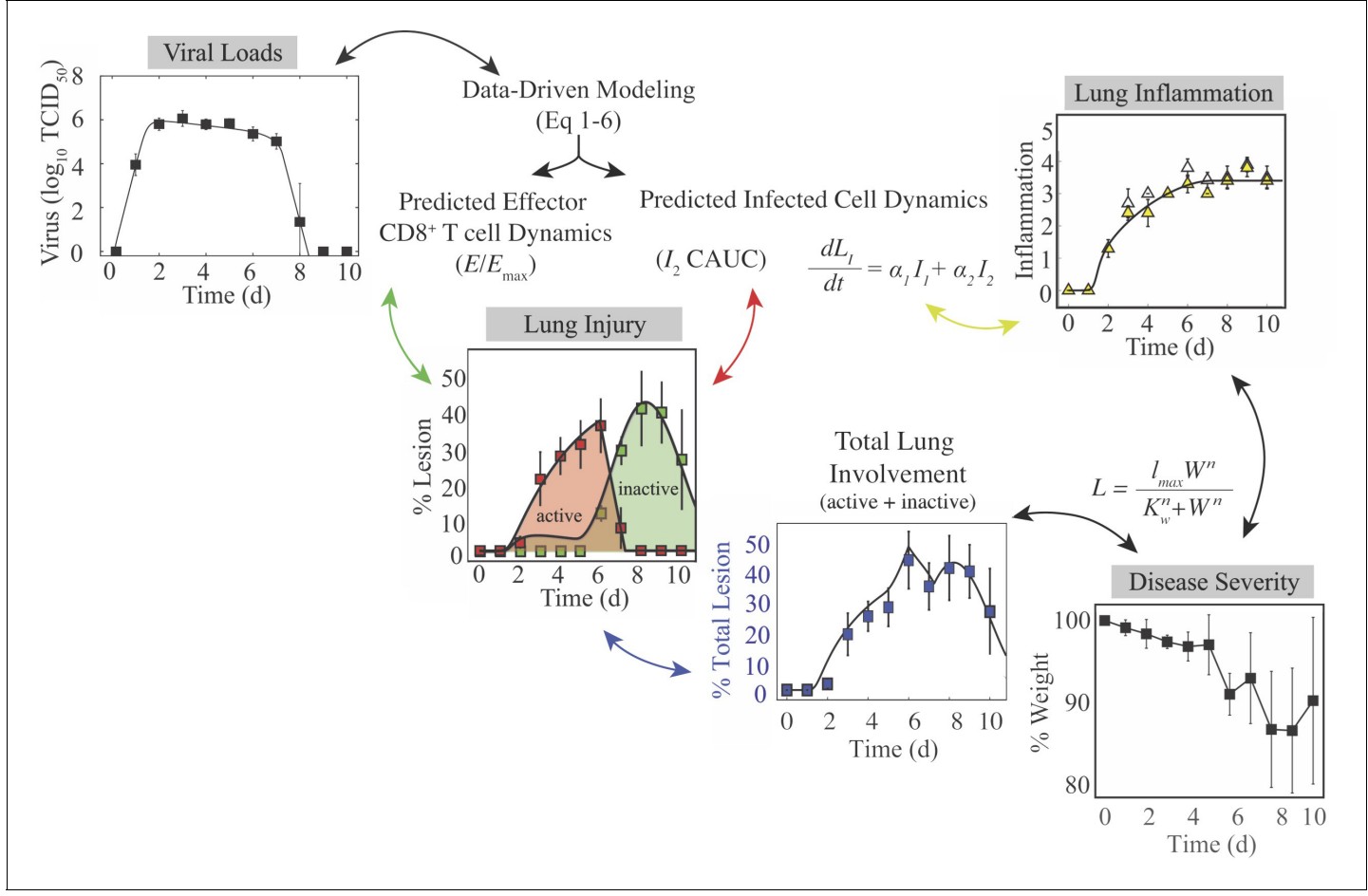

**Figure 6.** Summary of the connections between the kinetics of virus, infected cells, CD8$^+$ T cells, lung injury, lung inflammation, and disease severity. Summary of the relations between the dynamics of virus, infected cells, CD8$^+$ T cells, lung lesions, lung inflammation, and weight loss established by our analysis. Given that viral loads and weight loss are the most easily measured variables, our mathematical model (*Equation (1)-(6)*) could be used to estimate the kinetics of infected cells, CD8$^+$ T cells, and inflammation. The cumulative area under the curve (CAUC) of the productively infected cell dynamics ($I_2$) yields an estimate of the percent lung infected (active lesion) while the predicted relative CD8$_E$ dynamics ($E/E_{max}$) yields an estimate of the percent lung resolved (inactive lesion). The total amount of lung involved (% lung infected and % lung resolved) or the inflammation scores can then be used to estimate weight loss through the functions $L_L(W)$ and $L_I(W)$. These connections could be reversed and weight loss used to estimate viral load kinetics.

infection data and significant improvement in our ability to predict disease severity, the likelihood of complications, and therapeutic efficacy.

Our histomorphometric data and analyses provided a robust quantification of the extent of influenza virus infection within the lung and the dynamics of lung injury and inflammation. The images look strikingly similar to CT scans and postmortem histopathology from infected patients, which also show extensive bilateral damage and cytopathic effects in epithelial cells (*Kohr et al., 2010*; *Li et al., 2011*; *Ajlan et al., 2009*; *Soto-Abraham et al., 2009*; *Gill et al., 2010*; *Li et al., 2012*; *Shieh et al., 2010*). In addition, our results agree with weekly CT imaging of hospitalized patients infected with the 2009 H1N1 influenza virus showing that the damage worsens during the first week of illness and that cell proliferation begins in the second week (*Li et al., 2012*). Further investigating the utility of CT imaging could enhance the translatability of our findings. The model's 95% CI predictions of the lesioned area nicely captured the heterogeneity in the data (*Figure 4*), which is remarkable given the minimal variability in the virus and CD8$^+$ T cell data and model fit (*Figure 1*). The differential heterogeneity suggests that small changes to lung viral loads can induce significant changes in disease severity. Indeed, this has been observed in both humans and animals when antivirals are administered (*Toniolo Neto, 2014*; *Baloxavir Marboxil Investigators Group et al., 2018*; *Treanor et al., 2000*; *Moscona, 2005*; *Gubareva et al., 2000*; *Koshimichi et al., 2018*) or when host responses are experimentally modulated (*Channappanavar et al., 2016*; *Szretter et al., 2009*; *Iwasaki and Nozima, 1977*). It was also observed in our CD8 depletion data, where our model predicted a significant reduction in the lesioned area (*Figure 4*) while viral loads were only slightly lower (*Figure 3*). This may indicate that viral loads are generally not a reliable measure of infection severity, and the knowledge should aid future studies seeking to estimate therapeutic efficacy or the effects of immune modulation. There was a slight over-estimation of the inactive lesion early in the infection (1–5 d pi, dark gray shading; *Figure 4*) when using the fit to the total CD8$^+$ T cells, which was abrogated when using the CD25$^-$CD43$^+$CD8$^+$ T cells. However, these cells underestimated the total lesion at 6–7 d pi, suggesting contributions from other CD8$^+$ phenotypes. In general, modeling different CD8$^+$ T cell phenotypes may help clarify their *in vivo* functions and better define the rates of CD8-mediated infected cell clearance. Although our results suggest that this is not necessarily required, they may be more critical to consider when predicting the response of preexisting and potentially cross-reactive immunity particularly given that memory resident T cells are only beginning to be understood (*Zens et al., 2016*; *Slütter et al., 2017*; *Van Braeckel-Budimir et al., 2018*; *Wakim et al., 2015*; *Uddbäck et al., 2021*).

The new knowledge about the extent of infection within the lung also uncovered the nonlinear connection between disease severity and various measures of lung pathology (*Figure 5*). This discovery is significant because it suggests there is a unique link to the extent of both lung injury and inflammation, which are often used interchangeably yet have distinct dynamics (*Figure 4*) and correlate to different immune responses (*Figure 4—figure supplements 1–2*). In the same vein, both measurements can be estimated using the dynamics of infected cells (*Figure 4*) but with independent mathematical relations (CAUC of $I_2$ versus *Equation (7)*). Our CD8 depletion data (*Figure 3*) and other experimental studies using histomorphometry (*Marathe et al., 2016*) corroborate a relationship between weight loss and lung pathology during IAV infection. For example, animals treated with antivirals in various conditions (single or combination therapy and in immunocompetent or immunosuppressed hosts) demonstrated that, although viral loads are not always significantly reduced, the antiviral-induced reductions in weight loss were paired with decreased infected areas of the lung (*Marathe et al., 2016*). Further examining lung injury and inflammation kinetics and their connection to weight loss in various experimental infection settings (e.g., different ages or sex, under therapy, other strains or viruses, etc.) and deciphering how innate and humoral responses contribute to these measurements should improve our predictive capabilities. In addition, the nonlinearity of the connections warrants further investigation as it could indicate cooperativity of multiple mechanisms. Studies such as these may be particularly helpful in understanding the exacerbated morbidity and mortality in elderly individuals, where there are lower viral loads, increased weight loss, symptoms, and/or mortality within animal models (*Toapanta and Ross, 2009*; *Smith et al., 2019*). Because murine models are imperfect systems with important immunologic and physiologic distinctions from humans (*Mestas and Hughes, 2004*), uncovering how our results translate to human infection with novel or circulating influenza viruses and identifying the corresponding, tractable human symptom that could be a robust predictor of disease would be ideal. However, this could

be arduous due to inherent human and viral heterogeneity and contributions from co-morbidities and complex immunologic histories.

Despite their limitations, animal models are used to make viral and immunological discoveries and to develop and test therapeutics and vaccines (reviewed in *Thangavel and Bouvier, 2014*; *Margine and Krammer, 2014*; *Smith and McCullers, 2014*), and this work demonstrates a significant potential for the easily obtained weight loss data to be used, analyzed, and interpreted within both modeling and experimental studies. In our data and others' animal data, there is a spike in weight loss, symptom score, and/or inflammation at ~6 d pi (*Music et al., 2014*; *Kugel et al., 2009*; *Lu et al., 2018*; *Huang et al., 2017*; *Figure 5*). Our data suggests that this may be due to $CD8_E$ activity while the infection continues to spread. The increase in animal weights starting at 7 d pi is concurrent with the decline in infected lesions (*Figure 5*). Interestingly, there was no corresponding decrease in inflammation, and the spike was not apparent in the CD8 depletion data where severity was reduced. The subsequent increase in weight loss following infection resolution (8–9 d pi; *Figure 4C*) could be attributed to $CD8^+$ T cell-mediated pathology and/or to ongoing inflammation. The tighter correlation to inflammation might suggest the latter; however, some human and animal studies indicate that large numbers of $CD8^+$ T cells pose a risk of acute lung tissue injury (*Duan and Thomas, 2016*; *Moskophidis and Kioussis, 1998*; *Mauad et al., 2010*; *Rygiel et al., 2009*; *La Gruta et al., 2007*). According to our model predictions, excessive $CD8^+$ T cell numbers may augment disease progression yet do not improve recovery time (*Appendix 3—figure 2*). Instead, an earlier onset of $CD8_E$ proliferation (i.e., smaller $\tau_E$) would be required to significantly shorten the infection (*Appendix 3—figure 2*). This aligns with evidence that hosts with adaptive immune responses primed by vaccine or prior infection recover more rapidly (*Duan et al., 2015*; *Weinfurter et al., 2011*). While higher $CD8^+$ T cell numbers have little impact on viral kinetics, our model and data agree with clinical and experimental studies from a wide range of host species that impaired $CD8^+$ T cell responses can prolong an IAV infection (*Wang et al., 2015*; *Miao et al., 2010*; *McMichael et al., 1983*; *Yap and Ada, 1978*; *Wells et al., 1981*; *Bender et al., 1992*). In some scenarios, such as in immunocompromised hosts, virus can persist for up to several weeks if $CD8^+$ T cell-mediated clearance is unsuccessful (*Figure 3*; *Wang et al., 2015*; *Eichelberger et al., 1991*; *Hou et al., 1992*; *Xue et al., 2017*; *Memoli et al., 2014*). The bifurcation in recovery time revealed by the model suggests that this may occur when the number of lung-specific $CD8^+$ T cells are less than $2.31 \times 10^5$ cells (*Figure 3D–E*). Our CD8 depletion data show that the precise number will be dependent on other infection variables. Minimally, we would expect this number to vary depending on parameters like the dose, rate of virus replication, degree of prior immunity, and/or the infected cell number and lifespan, which has been noted in another modeling study that detailed similar bifurcating behavior (*Yates et al., 2011*). Although some previously published models also suggested delayed resolution with depleted $CD8^+$ T cell responses (*Dobrovolny et al., 2013*; *Miao et al., 2010*; *Cao et al., 2016*; *Lee et al., 2009*), this bifurcation has not been observed and their estimated delays in recovery do not amount to the long-lasting infections in immunodeficient patients (*Wang et al., 2015*; *Eichelberger et al., 1991*; *Hou et al., 1992*). Our model's ability to capture the dynamics when $CD8^+$ T cells are depleted is encouraging, and the data are an important reminder that experimental modulation of cell populations (e.g., through depletion or genetic knockouts) is complicated and that the data from such systems should be interpreted cautiously. It also brings into question studies that have used these types of data without validated mathematical models or quantification of other immunological variables.

Our data and analyses provided strong support that the infected cell density impacts the rate at which they are cleared by effector $CD8^+$ T cells ($CD8_E$) (*Figures 3–4*, *Figure 4—figure supplement 1*). Although $CD8^+$ T cell dynamics can be somewhat replicated when assuming the density dependence lies within their expansion, that assumption could not recover the viral load dynamics (*Appendix 1—figure 1*). Discriminating between these mechanisms is difficult *in vivo*, but the ability of the model in *Equation (1)-(6)* to capture the entire time course of $CD8_E$ dynamics in multiple scenarios is compelling. Regardless of the mechanism, we first detected this density-dependence in a model that excluded specific immune responses (see Appendix 2) (*Smith et al., 2018*). Simple models like that one are useful to capture nonlinearities, but they cannot distinguish between different mechanisms.

Several factors may contribute to the density-dependent change in the rate of $CD8_E$-mediated clearance. One possibility is that the slowed rate of clearance at high infected cell densities is due to

a 'handling time' effect, which represents the time required for an immune cell to remove an infected cell (e.g., as in *Smith et al., 2018*; *Li and Handel, 2014*; *Le et al., 2014*; *Graw and Regoes, 2009*; *Gadhamsetty et al., 2014*; *Pilyugin and Antia, 2000*; *Smith et al., 2011b*). When CD8$_E$ interact with infected cells, a complex is formed for ~20–40 min (*Mempel et al., 2006*; *Deguine et al., 2010*; *Zagury et al., 1975*; *Yannelli et al., 1986*; *Perelson and Bell, 1982*). Because CD8$_E$ could not interact with other infected cells during this time, the global rate of infected cell clearance would be lowest when infected cells outnumber CD8$_E$. In addition, contact by a single CD8$_E$ may be insufficient to remove an infected cell (*Halle et al., 2016*). Infected cell clearance is more frequently observed after interactions with multiple CD8$_E$, with an average of 3.9 contact events either serially or simultaneously (*Halle et al., 2016*). Thus, the high density of infected cells early in the infection reduces the probability that a single virus-infected cell would be targeted a sufficient number of times to induce cell death. However, as CD8$_E$ accumulate and the density of infected cells decreases (*Figure 3A*), the probability of simultaneous interactions will increase. These interactions may also be influenced by CD8$_E$ movement, where their velocity slows at the height of infected cell clearance (6–8 d pi) (*Lambert Emo et al., 2016*; *Matheu et al., 2013*). This should reduce the time required to remove an infected cell and, thus, result in a higher efficiency. Moreover, it is possible that spatial limitations also contribute, such that a high infected cell density may hinder CD8$^+$ T cells from reaching infected cells on the interior of the infected tissue. Crowding of immune cells at the periphery of infected lesions has been observed in other infections (e.g., in tuberculosis granulomas [*Ulrichs et al., 2004*; *Russell et al., 2009*]) and has been suggested in agent-based models of influenza virus infection (*Levin et al., 2016*).

In addition to illuminating the connections between viral kinetics and pathology, the histomorphometric data validated the model's infected cell dynamics (*Figure 4B*). The dynamics of susceptible and infected cells throughout the infection and the accuracy of the target cell limited approximation used within influenza viral kinetic models have been questioned for several years (*Smith and Perelson, 2011*; *Smith, 2018b*; *Beauchemin and Handel, 2011*; *Ahmed et al., 2017*; *Gallagher et al., 2018*; *Boianelli et al., 2015*; *Handel et al., 2018*). The ability of the model to accurately predict the histomorphometry and CD8$^+$ T cell depletion data corroborates the use of this approximation, which assumes a limited number of available target cells and describes their decline by infection only. Although the slowing of viral loads beginning at 2 d pi could be due to a variety of innate immune-mediated mechanisms (e.g., macrophages, neutrophils, type I interferons, etc.), adding more complex dynamics to our model was unnecessary to describe the data. Further, the model and data agree that there are few infected cells during the time when viral loads are growing most rapidly (0–2 d pi; *Figures 1B* and *4B*). We previously used this information to derive approximations for the model and gain deeper insight into how each parameter influences the kinetics (*Smith et al., 2010*), which has helped numerous studies interpret their results (*Miao et al., 2010*; *Smith et al., 2011a*; *Smith, 2018b*; *Holder and Beauchemin, 2011*). The data also supports the model's hypothesis that that there is minimal clearance of infected cells prior to CD8$_E$ expansion (*Figure 4*). The knowledge of the model's accuracy and of the spatiality in the lung should aid investigation into the mechanisms that limit virus growth during the early stages of the infection.

Employing targeted model-driven experimental designs to examine and validate theoretical predictions like the ones presented here is pivotal to elucidating the mechanisms of infection spread and clearance (*Smith, 2018b*). Examining other infections (e.g., coronaviruses), modifications to the dynamics (e.g., lethal doses), and the connection between lung measurements and those more easily acquired from upper respiratory tract will help refine the dynamical links between virus, host immune responses, and disease severity and identify their generalizability. Determining the factors that influence disease severity is vital to understanding the disproportionate mortality in at-risk populations (e.g., elderly) and to improving therapeutic design. This is particularly important because current antivirals alleviate symptoms but do not always effectively lower viral loads (*Toniolo Neto, 2014*; *Baloxavir Marboxil Investigators Group et al., 2018*; *Treanor et al., 2000*; *Moscona, 2005*; *Gubareva et al., 2000*; *Koshimichi et al., 2018*). The predictive capabilities of validated models like the one here should prove useful in forecasting infection dynamics for a variety of scenarios. These tools and analyses provide a more meaningful interpretation of infection data, new ways to interpret weight loss data in animal models, and a deeper understanding of the progression and resolution of the disease, which will undoubtedly aid our ability to effectively combat influenza.

## Materials and methods

### Ethics statement

All experimental procedures were performed under protocols O2A-020 or 17–096 approved by the Animal Care and Use Committees at St. Jude Children's Research Hospital (SJCRH) or the University of Tennessee Health Science Center (UTHSC), respectively, under relevant institutional and American Veterinary Medical Association (AVMA) guidelines. All experimental procedures were performed in a biosafety level two facility that is accredited by the American Association for Laboratory Animal Science (AALAS).

### Mice

Adult (6-week-old) female BALB/cJ mice were obtained from Jackson Laboratories (Bar Harbor, ME) or Charles River Laboratories (Wilmington, Massachusetts). Mice were housed in groups of five mice in high-temperature 31.2 cm × 23.5 cm × 15.2 cm polycarbonate cages with isolator lids (SJCRH) or in 38.2 cm × 19.4 cm × 13.0 cm solid-bottom polysulfone individually ventilated cages (UTHSC). Rooms used for housing mice were maintained on a 12:12 hr light:dark cycle at 22 ± 2°C with 50% humidity in the biosafety level two facility at SJCRH (Memphis, TN) or UTHSC (Memphis, TN). Prior to inclusion in the experiments, mice were allowed at least 7 days to acclimate to the animal facility such that they were 7 weeks old at the time of infection. Laboratory Autoclavable Rodent Diet (PMI Nutrition International, St. Louis, MO; SJCRH) or Teklad LM-485 Mouse/Rat Sterilizable Diet (Envigo, Indianapolis, IN; UTHSC) and autoclaved water were available *ad libitum*. All experiments were performed under an approved protocol and in accordance with the guidelines set forth by the Animal Care and Use Committee at SJCRH or UTHSC.

### Infectious agents

All experiments were done using the mouse adapted influenza A/Puerto Rico/8/34 (H1N1) (PR8).

### Infection experiments

The viral infectious dose ($TCID_{50}$) was determined by interpolation using the method of *Reed and Muench, 1938* using serial dilutions of virus on Madin-Darby canine kidney (MDCK) cells. Mice were intranasally inoculated with 75 $TCID_{50}$ PR8 diluted in 100 μl of sterile PBS. In a subset of animals, $CD8^+$ T cells were depleted by IP injection at days −2, 0, 3, and 7 pi with 100 μg of the rat anti-$CD8\alpha$ antibody (clone 2.43) that was purified from ATCC hybridoma (per manufacturer instructions) in 250 μl of PBS. Depletion efficiency was confirmed in the lung and spleen as described below. Mice were weighed at the onset of infection and each subsequent day to monitor illness and mortality. Mice were euthanized if they became moribund or lost 30% of their starting body weight. For viral load and total $CD8^+$ T cell quantification, experiments were repeated three times and in each facility to ensure reproducibility and two complete experiments (ten animals per time point) were used for these studies. For all other experiments, the data was compared to prior results in addition to being repeated at select time points to ensure reproducibility, and one complete experiment (five animals per time point) was used. For pathology scoring and histomorphometry quantitation, five animals per time point were used. Power was calculated using G*Power.

### Lung harvesting for viral and cellular dynamics

For total $CD8^+$ T cell quantification, mice were euthanized by $CO_2$ asphyxiation. To distinguish blood-borne $CD8^+$ T cells from those in the lung parenchyma, deeply anesthetized mice (5% isoflurane) were retro-orbitally injected with 3 μg of anti-CD45 antibody (PerCP, clone 30-F11, Biolegend) 3 min prior to euthanasia (*Anderson et al., 2012*; *Keating et al., 2018*). Mice were then euthanized by 33% isoflurane inhalation and their lungs perfused with 10 ml PBS prior to removal. For all experiments, lungs were then aseptically harvested, washed three times in PBS, and placed in 500 μl sterile PBS. Whole lungs were digested with collagenase (1 mg/ml, Sigma C0130), and physically homogenized by syringe plunger against a 40 μm cell strainer. Cell suspensions were centrifuged at 4°C, 500xg for 7 min. The supernatants were used to determine the viral titers ($TCID_{50}$) by serial dilutions on MDCK monolayers. Following red blood cell lysis, cells were washed in MACS buffer (PBS, 0.1M EDTA, 0.01M HEPES, 5 mM EDTA and 5% heat-inactivated FBS). Cells were then counted with

trypan blue exclusion using a Cell Countess System (Invitrogen, Grand Island, NY) and prepared for flow cytometric analysis as indicated below.

## Lung titers

For each animal, viral titers were obtained using serial dilutions on MDCK monolayers and normalized to the total volume of lung homogenate supernatant. The resulting viral loads are shown in *Figure 1B*, provided in *Source data 1*, and were previously published and utilized for calibration of the density-dependent model (see Appendix 2; *Smith et al., 2018*).

## Flow cytometric analysis

Flow cytometry (LSRII, BD Biosciences, San Jose, CA (SJCRH) or ZE5 Cell Analyzer, Bio-Rad, Hercules, CA (UTHSC)) was performed on the cell pellets after incubation with 200 µl of a 1:2 dilution of Fc block (human-$\gamma$ globulin) on ice for 30 min, followed by viability (Biolegend, Zombie Violet Fixable Viability Kit) and surface marker staining with anti-mouse antibodies. For total $CD8^+$ T cell, macrophage, and neutrophil quantification, we used antibodies CD3e (BV786, clone 145–2 C11, Biolegend), CD4 (PE-Cy5, clone RM4-5, BD Biosciences), CD8$\alpha$ (BV605, clone 53–6.7, BD Biosciences), Ly6C (APC, clone HK1.4, eBioscience), F4/80 (PE, clone BM8, eBioscience), CD11c (eFluor450, clone N418, eBioscience), CD11b (Alexa700, clone M1/70, BD Biosciences), MHC-II (FITC, clone M5/114.15.2, Invitrogen), CD49b (APCe780, clone DX5, eBioscience), and Ly6G (PerCp-Cy5.5, clone 1A8, Biolegend). The data were analyzed using FlowJo 10.4.2 (Tree Star, Ashland, OR) where viable cells were gated from a forward scatter/side scatter plot and singlet inclusion. Following neutrophil ($Ly6G^{hi}$) and subsequent macrophage ($CD11c^{hi}F4/80^{hi}$) exclusion, $CD8^+$ T cells were gated as $CD3e^+DX5^-CD4^-CD8^+$.

To distinguish blood borne $CD8^+$ T cells from those in the lung parenchyma, we used antibodies CD3e (BV786, clone 145–2 C11, Biolegend), CD4 (FITC, clone RM4-5, Biolegend), CD8$\alpha$ (BV605, clone 53–6.7, BD Biosciences), B220 (APCe780, clone RA3-6B2, eBioscience), CD49b (APCe780, clone DX5, eBioscience), CD62L (PE-Cy7, clone MEL-14, Biolegend), CD69 (PE, clone H1.2F3, Biolegend), CD44 (PE-Dazzle594, clone IM7, Biolegend), CD25 (Alexa700, clone PC61, Biolegend), CD43 (APC, clone 1B11, Biolegend), CD127 (PE-Cy5, clone A7R34, Biolegend), and CD103 (BV711, clone 2E7, Biolegend) or CD314 (NKG2D) (BV711, clone CX5, BD Bioscience). The data were analyzed using FlowJo 10.6.2 (Tree Star, Ashland, OR) where viable cells were gated from a forward scatter/side scatter plot, singlet inclusion, and viability dye exclusion. Blood borne $CD8^+$ T cells were gated as $CD45^+$ and those in the lung parenchyma as $CD45^-$. The total in each sub-population was gated as $CD3e^+B220^-DX5^-CD4^-CD8^+$. IAV-specific $CD8^+$ T cells were then gated as $CD25^+CD43^+$ (recently activated), $CD25^-CD43^+$ (effector) (*Keating et al., 2018*), and $CD43^-CD127^+CD103^+CD69^+$ (long-lived memory). Expression of CD44, CD69, CD62L, and NKG2D were also assessed to ensure appropriate classification.

For all experiments, the absolute numbers of $CD8^+$ T cells were calculated based on viable events analyzed by flow cytometry as related to the total number of viable cells per sample. The data are provided in *Source data 1*. We use the kinetics of the total number of $CD8^+$ T cells (non-perfused lung; 'total'), the total number of $CD8^+$ T cells in the lung parenchyma (perfused lung; 'lung-specific'), and virus-specific $CD8^+$ T cell phenotypes in the lung parenchyma as defined by surface staining (described above). We chose this approach because the use of tetramers yields epitope-specific cell dynamics that vary in time and magnitude (e.g., as in *Toapanta and Ross, 2009*; *Keating et al., 2018*) and would complicate the model and potentially skew the results. In addition, tetramer staining is not available for all epitopes in BALB/cJ mice and they may underestimate the dynamics of IAV-specific cells (*Keating et al., 2018*). The gating strategies are shown in *Figure 1—figure supplement 2*.

## Lung immunohistopathologic and immunohistochemical (IHC) evaluation

The lungs from IAV-infected mice were fixed via intratracheal infusion and then immersion in 10% buffered formalin solution. Tissues were paraffin embedded, sectioned, and stained for influenza virus using a primary goat polyclonal antibody (US Biological, Swampscott, MA) against influenza A, USSR (H1N1) at 1:1000 and a secondary biotinylated donkey anti-goat antibody (sc-2042; Santa Cruz Biotechnology, Santa Cruz, CA) at 1:200 on tissue sections subjected to antigen retrieval for 30 min

at 98˚C. The extent of virus spread was quantified by capturing digital images of whole-lung sections (two dimensional) stained for viral antigen using an Aperio ScanScope XT Slide Scanner (Aperio Technologies, Vista, CA) then manually outlining defined fields. Alveolar areas containing virus antigen-positive pneumocytes were highlighted in red (defined as 'active' infection), whereas lesioned areas containing minimal or no virus antigen-positive debris were highlighted in green (defined as 'inactive' infection). Lesions containing a mix of virus antigen-positive and antigen-negative pneumocytes were highlighted in orange (defined as 'mixed' infection). The percentage of each defined lung field was calculated using the Aperio ImageScope software. Pulmonary lesions in HE-stained histologic sections were assigned scores based on their severity and extent. Representative images from each group and quantitative analyses (five animals/group) of viral spread and lung pathology during IAV infection are shown in *Figure 4*, and provided in *Source data 1*.

## CD8$^+$ T cell model

To examine the contribution of CD8$^+$ T cells to the biphasic viral load decay, we expanded the density-dependent (DD) model (see Appendix 2) to include two mechanisms of infected cell clearance (non-specific clearance ($\delta$) and CD8$^+$ T cell-mediated clearance ($\delta_E(I_2, E)$)) and two CD8$^+$ T cell populations: effector ($E$, denoted CD8$_E$) and memory ($E_M$, denoted CD8$_M$) CD8$^+$ T cells. The model is given by *Equation (1)-(6)*.

$$\frac{dT}{dt} = -\beta TV \tag{1}$$

$$\frac{dI_1}{dt} = \beta TV - kI_1 \tag{2}$$

$$\frac{dI_2}{dt} = kI_1 - \delta I_2 - \delta_E(I_2, E)I_2 \tag{3}$$

$$\frac{dV}{dt} = pI_2 - cV \tag{4}$$

$$\frac{dE}{dt} = \xi(E)I_2 + \eta EI_2(t - \tau_E) - d_E E \tag{5}$$

$$\frac{dE_M}{dt} = \zeta E(t - \tau_M) \tag{6}$$

In this model, target cells become infected with virus at rate $\beta V$ per day. Once infected, cells enter an eclipse phase ($I_1$) before transitioning at rate $k$ per day to a productively-infected state ($I_2$). These infected cells produce virus at rate $p$ TCID$_{50}$ per infected cell per day, and virus is cleared at rate $c$ per day. Virus-producing infected cells ($I_2$) are cleared by non-specific mechanisms (e.g., apoptosis and/or innate immune responses) at a constant rate $\delta$ per day. Innate immune responses were excluded from the model because the viral load data is linear during the time where they act (2–7 d pi) and, thus, additional equations cannot improve the fit. Productively infected cells are cleared by CD8$_E$ at rate $\delta_E(I_2, E) = \delta_E E/(K_{\delta_E} + I_2)$ per day, where the rate of infected cell clearance is $\delta_E/K_{\delta_E}$ per CD8$_E$ per day and $K_{\delta_E}$ is the half-saturation constant. The CD8$_E$-mediated clearance rate ($\delta_E(I_2, E)$) is dependent on the density of infected cells and is similar to the infected cell clearance term in the DD model (see *Equation (A11)*; *Smith et al., 2018*). Similar density-dependent forms have also been used in models that describe the CD8$^+$ T cell response to other virus infections (*De Boer and Perelson, 1998*; *Li and Handel, 2014*; *Gadhamsetty et al., 2014*). Models that exclude this density-dependence were examined, but these models resulted in a statistically poor fit to the data as defined by the Akaike Information Criteria (AIC) (*Supplementary file 2*). This is due in part to the increase of CD8$^+$ T cells from 3-5 d pi (*Figure 1*). We did examine a model that excluded these dynamics and included CD8$^+$ T cell expansion as a density-dependent function (e.g., $\eta EI_2(t - \tau_E)/(K_I + I_2)$) while keeping a linear rate of CD8-mediated infected cell clearance ($\delta_E EI_2$) (see Appendix 1). This model was adapted from other published models (*Conway and Perelson,*

*2015*; *Bonhoeffer et al., 2000*; *Baral et al., 2019*) and can produce similar dynamics from 6-10 d pi, but it was not statistically supported by our data (*Supplementary file 1*). We further tested the model in *Baral et al., 2019*, but this model could not fit our data and lacked statistical support (*Supplementary file 1*).

The model assumes that the initial CD8$_E$ influx in the lung is proportional to infected cells at rate $\xi(E) = \xi/(K_E + E)$ CD8$_E$ per cell per day, which is down-regulated by the CD8$_E$ already present in the lung. The associated half-saturation constant is $K_E$. Similar terms for CD8$_E$ regulation have been used in modeling HIV infections (*De Boer and Perelson, 1998*; *Müller et al., 2001*) and in models that examined CD8$^+$ T cell proliferation mechanisms (*De Boer and Perelson, 1995*). We also examined whether their influx is proportional to infected cells at a delay (i.e., $\xi(E)I(t - \tau_I)$). While this modification better captured the initial increase in CD8$^+$ T cells, the additional parameter was not supported. In our model, CD8$_E$ expansion occurs at rate $\eta$ per infected cell per day with time delay $\tau_E$. This term accounts for local CD8$_E$ proliferation in the lung (*McGill and Legge, 2009*; *Lawrence et al., 2005*) and migration of CD8$_E$ from secondary lymphoid organs (*Zhang and Bevan, 2011*; *Bedoui and Gebhardt, 2011*; *Wu et al., 2011*; *Kang et al., 2011*). The delay may signify the time it takes CD8$_E$ to become activated by antigen presenting cells, differentiate, proliferate, and/or migrate to the infection site. The lung CD8$_E$ population declines due to cell death and/or emigration at rate $d_E$ per day. These cells transition to CD8$_M$ ($E_M$) at rate $\zeta$ CD8$_M$ per CD8$_E$ per day after $\tau_M$ days. The model schematic and fit to the viral load, total, effector, and memory CD8$^+$ T cell data are in *Figure 1*. All code is provided in *Source code 1*.

## Inflammation model

To estimate the alveolar and interstitial inflammation without modeling other cell classes (e.g., macrophages and neutrophils), we assumed that inflammation in the lung ($L_I$) was proportional to the infected cells ($I_1$ and $I_2$) according to the equation,

$$\frac{dL_I}{dt} = \alpha_1 I_1 + \alpha_2 I_2, \qquad (7)$$

where $\alpha_{1,2}$ has units of score/cell/d and defines the inflammation score contribution from each infected cell class. A decay term was excluded because inflammation does not resolve on the time-scale of our data.

## Parameter estimation

Given a parameter set $\theta$, the cost $C(\theta)$ was minimized across parameter ranges using an Adaptive Simulated Annealing (ASA) global optimization algorithm (*Smith et al., 2018*) to compare experimental and predicted values of virus ($V$; log$_{10}$ TCID$_{50}$/lung) and log$_{10}$ total CD8$^+$ T cells/lung ($\hat{E} = E + E_M + \hat{E}_0$, where $\hat{E}_0$ is the initial number of CD8$^+$ T cells at 0 d pi), or log$_{10}$ TCID$_{50}$/lung virus ($V$), log$_{10}$ effector CD8$^+$ T cells/lung ($\hat{E} = E + \hat{E}_0$), and log$_{10}$ memory CD8$^+$ T cells/lung ($\hat{E}_M = E_M + \hat{E}_{M_0}$). The cost function is defined by,

$$C(\theta) = \sum_{i,j}(V(\theta,t_i) - v_{i,j})^2 + \sum_{i,j}(\hat{E}_M(\theta,t_i) - m_{i,j})^2 + s_E\left[\sum_{i,j}(\hat{E}(\theta,t_i) - e_{i,j})^2 + \right.$$
$$\left. \sum_i \sqrt{\gamma_i}\left(\frac{\hat{E}(\theta,t_{i+1}) - \hat{E}(\theta,t_{i-1})}{t_{i+1} - t_{i-1}} - \frac{1}{\gamma_i}\sum_j \frac{e_{i+1,j} - e_{i-1,j}}{t_{i+1} - t_{i-1}}\right)^2\right],$$

where $(t_i, v_{i,j})$ is the viral load data, $(t_i, e_{i,j})$ is the total or effector CD8$^+$ T cell data, $(t_i, m_{i,j})$ is the memory CD8$^+$ T cell data, and $V(\theta,t_i)$, $\hat{E}(\theta,t_i)$, and $\hat{E_M}(\theta,t_i)$ are the corresponding model predictions. Here, $s_E = (v_{\max} - v_{\min})/(e_{\max} - e_{\min})$ is a scaling factor, and $\gamma_i = J_{i+1}J_{i-1}$ where $J_i$ is the number of observations at time $t_i$. Errors of the log$_{10}$ data were assumed to be normally distributed. To explore and visualize the regions of parameters consistent with the model, we fit *Equation (1)-(6)* to 2000 bootstrap replicates of the data. If the fit was within $\chi^2 = 0.05$ of the best-fit and the CD8 derivative was not a statistical outlier as determined by the function *isoutlier*, then the bootstrap was considered successful (*Smith et al., 2011a*; *Smith et al., 2013*; *Smith et al., 2018*). For each best-fit estimate, we provide 95% confidence intervals (CI) obtained from the bootstrap replicates (*Table 1*).

Calculations were performed either in MATLAB using a custom built ASA algorithm (*Smith et al., 2018*) or in Python using the *simanneal* package (*Perry, 2018*) followed by a L-BFGS-B (*Byrd et al., 1995*; *Zhu et al., 1997*) deterministic minimization through SciPy's *minimize* function. MATLAB *ode15s* and *dde23* or SciPy integrate.ode using *lsoda* and PyDDE (*Cairns, 2008*) were used as the ODE and DDE solvers.

Estimated parameters in the CD8$^+$ T cell model included the rates of virus infection ($\beta$), virus production ($p$), virus clearance ($c$), eclipse phase transition ($k$), non-specific infected cell clearance ($\delta$), CD8$_E$-mediated infected cell clearance ($\delta_E$), half-saturation constants ($K_{\delta_E}$ and $K_E$), CD8$_E$ infiltration ($\xi$), CD8$_E$ expansion ($\eta$), delay in CD8$_E$ expansion ($\tau_E$), CD8$_E$ clearance ($d_E$), CD8$_M$ generation ($\zeta$), delay in CD8$_M$ generation ($\tau_M$), and the baseline number of CD8$^+$ T cells ($\hat{E}_0$, $\hat{M}_0$). Bounds were placed on the parameters to constrain them to physically realistic values. Because biological estimates are not available for all parameters, ranges were set reasonably large based on preliminary results and previous estimates (*Smith et al., 2018*). The rate of infection ($\beta$) was allowed to vary between $10^{-6} - 10^{-1}$ TCID$_{50}^{-1}$ d$^{-1}$, and the rate of virus production ($p$) between $10^{-1} - 10^3$ TCID$_{50}$ cell$^{-1}$ d$^{-1}$. Bounds for the virus clearance rate ($c$) were 1 d$^{-1}$ ($t_{1/2} = 16.7$ h) and $10^3$ d$^{-1}$ ($t_{1/2} = 1$ min). To ensure biological feasibility, the lower and upper bounds for the eclipse phase transition rate ($k$) were 4-6 d$^{-1}$ as done previously (*Smith et al., 2018*).

The rate of non-specific infected cell clearance ($\delta$) was given limits of 0.05–1 d$^{-1}$. The CD8$_E$-mediated infected cell clearance rate ($\delta_E$) varied between $0.01 - 2$ cells CD8$_E^{-1}$ d$^{-1}$, and the associated half-saturation constant ($K_{\delta_E}$) was bounded between $10^1$–$10^6$ cells. The upper bound of $\delta_E$ was chosen to maintain the convergence of $\delta$ to nonzero values and consistency with prior results where $\delta$ approximates the slope of the first decay phase (*Smith et al., 2010*). Bounds for the rate of CD8$_E$ infiltration ($\xi$) were $10^2 - 10^6$ CD8$_E^2$ cell$^{-1}$ d$^{-1}$, and bounds for the half-saturation constant ($K_E$) were $10^3$-$10^7$ CD8$_E$. The CD8$_E$ expansion rate ($\eta$) varied between $10^{-8}$–$10^{-6}$ cell$^{-1}$ d$^{-1}$, and the delay in CD8$_E$ expansion ($\tau_E$) between 2–6 d. The rate of CD8$_E$ clearance ($d_E$) had limits of 0.05–2 d$^{-1}$. The rate of CD8$_M$ generation ($\zeta$) varied between $0.01 - 1$ CD8$_M$ CD8$_E^{-1}$ d$^{-1}$, and the delay in CD8$_M$ generation ($\tau_M$) varied between 3–4 d (total CD8$^+$ T cell fit) or 2–6 d pi (effector and memory CD8$^+$ T cell fit). Larger bounds were examined for this parameter; however, the parameter was non-identifiable in the total CD8$^+$ T cell fit and a small range was required for convergence. Bounds for the baseline number of CD8$^+$ T cells ($\hat{E}_0$, $\hat{E}_{M_0}$) were set to the upper and lower values of the data at 0 d pi ($3.0 \times 10^5 - 5.3 \times 10^5$ CD8 (total), $2.5 \times 10^2 - 1.4 \times 10^3$ CD8$_E$ (effector), $3.4 \times 10^2 - 8.0 \times 10^2$ CD8$_M$ (memory)).

The initial number of target cells ($T(0)$) was set to $10^7$ cells (*Smith et al., 2011a*). The initial number of infected cells $I_1(0)$ was set to 75 cells to reflect an initial dose of 75 TCID$_{50}$ (*Smith et al., 2018*). We previously found that estimating $I_1(0)$, fixing $V(0) = 75$ TCID$_{50}$, or estimating $V(0)$ did not improve the fit and could not be statistically justified (*Smith et al., 2018*). The initial number of productively infected cells ($I_2(0)$), the initial free virus ($V(0)$), and the initial number of CD8$_E$ ($E(0)$) and CD8$_M$ ($E_M(0)$) were set to 0. All other parameter estimations were done as described in the text.

## Linear regression

The function *polyfit* in MATLAB was used to perform linear regression on the percent active lesioned area, the percent inactive lesioned area, and the CD8$^+$ T cells during the expansion phase (5–8 d pi) and the contraction phase (9–10 d pi). Linear fits are shown in *Figure 4—figure supplement 1*.

## Cumulative area under the curve

The function *cumtrapz* in MATLAB was used to estimate the cumulative area under the curve (CAUC) of the infected cells ($I_2$) for the best-fit model solution.

# Acknowledgements

This work was supported by NIH grants AI100946, AI125324, and AI139088 and ALSAC. We thank Alan Perelson for his helpful comments, and Robert Michael for technical assistance.

## Additional information

### Funding

| Funder | Grant reference number | Author |
|---|---|---|
| National Institute of Allergy and Infectious Diseases | AI139088 | Margaret A Myers<br>Amanda P Smith<br>Lindey C Lane<br>Rosemary Aogo |
| National Institute of Allergy and Infectious Diseases | AI125324 | Margaret A Myers<br>Amanda P Smith<br>Lindey C Lane<br>David J Moquin<br>Amber M Smith |
| National Institute of Allergy and Infectious Diseases | AI100946 | Amber M Smith |
| American Lebanese Syrian Associated Charities | Internal Funding | Margaret A Myers<br>Amanda P Smith<br>Lindey C Lane<br>David J Moquin<br>Stacie Woolard<br>Paul Thomas<br>Peter Vogel<br>Amber M Smith |

The funders had no role in study design, data collection and interpretation, or the decision to submit the work for publication.

### Author contributions

Margaret A Myers, Software, Formal analysis, Investigation, Visualization, Writing - original draft; Amanda P Smith, Data curation, Formal analysis, Validation, Investigation, Visualization, Methodology, Writing - original draft, Writing - review and editing; Lindey C Lane, David J Moquin, Data curation; Rosemary Aogo, Formal analysis, Writing - review and editing; Stacie Woolard, Formal analysis; Paul Thomas, Methodology, Writing - review and editing; Peter Vogel, Data curation, Formal analysis, Visualization, Methodology, Writing - original draft, Writing - review and editing; Amber M Smith, Conceptualization, Resources, Data curation, Software, Formal analysis, Supervision, Funding acquisition, Validation, Investigation, Visualization, Methodology, Writing - original draft, Writing - review and editing

### Author ORCIDs

Amber M Smith ⓘD https://orcid.org/0000-0002-7092-6904

### Ethics

Animal experimentation: All experimental procedures were performed under protocols O2A-020 or 17-096 approved by the Animal Care and Use Committees at St. Jude Children's Research Hospital (SJCRH) or the University of Tennessee Health Science Center (UTHSC), respectively, under relevant institutional and American Veterinary Medical Association (AVMA) guidelines. All experimental procedures were performed in a biosafety level 2 facility that is accredited by the American Association for Laboratory Animal Science (AALAS).

### Decision letter and Author response

Decision letter https://doi.org/10.7554/eLife.68864.sa1
Author response https://doi.org/10.7554/eLife.68864.sa2

# Additional files

## Supplementary files

• Source data 1. Source data including viral titers, CD8$^+$ T cells, inflammation, lesion, and weight loss.

• Source code 1. Source code files for reproduction of all figures.

• Supplementary file 1. Statistical comparison of alternate models. Comparison of the Akaike Information Criteria (AIC) of the CD8$^+$ T cell model in *Equation (1)-(6)*, the alternate model in *Equation (A1)-(A6)*, and the Baral model in *Equation (A7)-(A8)*. The fit of these models is shown in *Appendix 1—figure 1*.

• Supplementary file 2. CD8 T cell depletion model parameters. Parameters, SSR, and AIC$_C$ obtained from fitting the CD8$^+$ T cell model (*Equation (1)-(6)*) to viral titers and CD8$^+$ T cells from mice infected with 75 TCID$_{50}$ PR8 and with CD8$^+$ T cells depleted at $-2$ d, 0 d, 3 d, and 7 d pi. The total number of CD8$^+$ T cells is $\hat{E} = E + E_M + \hat{E}_0$, where $\hat{E}_0 = 5.6 \times 10^3$ cells, and all other parameters are those in *Table 1*. The best model is bolded.

• Transparent reporting form

## Data availability

All data generated or analyzed during this study are included in the manuscript and supporting files. Source data files have been provided.

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

## Appendix 1

## Alternate CD8$^+$ T cell Models

We examined alternate formulations of the CD8$^+$ T cell model to further investigate the density-dependence in the CD8$^+$ T cell response. Rather than assuming that CD8$_E$-mediated clearance of infected cells is dependent on their density, the model in *Equation (A1)-(A6)* assumes that the rate of CD8$_E$ expansion is dependent on the density of infected cells. Similar models have been used to study CD8$^+$ T cell responses during HIV infection (*Conway and Perelson, 2015*; *Bonhoeffer et al., 2000*) and other viral infections (*Baral et al., 2019*). The differences between this alternate model and the CD8$^+$ T cell model (*Equation (1)-(6)*) are in bold.

$$\frac{dT}{dt} = -\beta TV \tag{A1}$$

$$\frac{dI_1}{dt} = \beta TV - kI_1 \tag{A2}$$

$$\frac{dI_2}{dt} = kI_1 - \delta I_2 - \mathbf{\delta_{Ea}EI_2} \tag{A3}$$

$$\frac{dV}{dt} = pI_2 - cV \tag{A4}$$

$$\frac{dE}{dt} = \frac{\mathbf{\eta_{Ea}}}{\mathbf{K_{Ea}} + \mathbf{I_2}} I_2(t - \tau_E) - d_E E \tag{A5}$$

$$\frac{dE_M}{dt} = \zeta E(t - \tau_M) \tag{A6}$$

When a linear CD8$_E$-mediated infected cell clearance rate is included, the CD8$_E$ dynamics between 3–5 d pi cannot be replicated and returned a statistically worse fit via AIC (*Supplementary file 1*). However, because these cells may not have effector functions and contribute to infected cell clearance at this time (see Discussion), we excluded these data and the term $\xi I_2/(K_E + E)$ when fitting *Equation (A1)-(A6)* to the viral load and total CD8$^+$ T cell data (*Appendix 1—figure 1*). The CD8 dynamics are similar to those generated by CD8$^+$ T cell model. However, the alternate model underestimates the data at day seven and the sharp decline between 7–8 d. While it is inappropriate to directly compare these models due to the varying number of data points, we assessed their goodness of fit with and without contributions from the data at 3–5 d pi (*Supplementary file 1*). Regardless of the data inclusion or exclusion, the alternate model had a higher AIC in all contexts and, thus, is not statistically justifiable. Similarly, we examined the model in *Baral et al., 2019* (*Equation (A7)-(A8)*), which also could not replicate our data (*Appendix 1—figure 1*).

$$\frac{dI}{dt} = \kappa_b I \left(1 - \frac{I}{I_{max}}\right) - \delta_{Eb} IE \tag{A7}$$

$$\frac{dE}{dt} = \eta_b \frac{IE}{K_{Eb} + I} - d_{Eb} \frac{IE}{K_{Db} + I} \tag{A8}$$

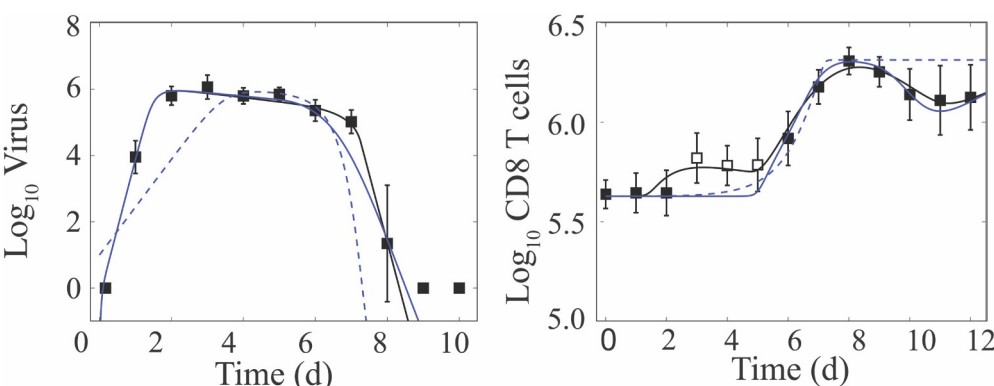

**Appendix 1—figure 1.** Fit of alternate CD8+ T cell models. Fit of the CD8+ T cell model (*Equation (1)-(6)*; solid black line) compared to the fit of two alternate CD8+ T cell models (*Equation (A1)-(A6)* (solid blue line) and *Equation (A7)-(A8)* (dashed blue line)) to virus and CD8+ T cells (excluding 3–5 d pi; white squares) from the lungs of mice infected with 75 TCID$_{50}$ PR8 (10 mice per time point). Resulting parameter values were $\delta_{Ea} = 4.02 \times 10^{-6}$ CD8$_{\mathrm{E}}^{-1}$d$^{-1}$, $\eta_{Ea} = 3.12 \times 10^{5}$ CD8$_{\mathrm{E}}$/d, and $K_{Ea} = 9.53 \times 10^{5}$ infected cells, and $\kappa_b = 3.33$ d$^{-1}$, $I_{max} = 1.0 \times 10^{6}$ infected cells, $\delta_{Eb} = 1.53 \times 10^{-5}$ CD8$_{\mathrm{E}}^{-1}$d$^{-1}$, $\eta_b = 1.49$ d$^{-1}$, $K_{Eb} = 9.5$ infected cells, $d_{Eb} = 0.95$ d$^{-1}$, and $K_{Db} = 1.90 \times 10^{6}$ infected cells. All other parameters are in *Table 1*, and the AICs are in *Supplementary file 1*. Data are shown as mean ± standard deviation.

## Appendix 2

### Comparison of the Density-Dependent and CD8$^+$ T cell Models

We previously developed a density-dependent (DD) viral kinetic model, which describes the biphasic decline of viral loads without inclusion of specific host responses (**Smith et al., 2018**). This model tracks four populations: susceptible epithelial ('target') cells ($T$), two classes of infected cells ($I_1$ and $I_2$), and virus ($V$) (**Smith et al., 2018**).

$$\frac{dT}{dt} = -\beta TV \tag{A9}$$

$$\frac{dI_1}{dt} = \beta TV - kI_1 \tag{A10}$$

$$\frac{dI_2}{dt} = kI_1 - \delta_d(I_2)I_2 \tag{A11}$$

$$\frac{dV}{dt} = pI_2 - cV \tag{A12}$$

Briefly, in the DD model, virus-producing infected cells ($I_2$) are cleared according to the function $\delta_d(I_2) = \delta_d/(K_\delta + I_2)$, where $\delta_d/K_\delta$ is the maximum per day rate of infected cell clearance and $K_\delta$ is the half-saturation constant (**Appendix 2—figure 1**). All other terms are common to the CD8 T$^+$ cell model (**Equation (1)-(6)**). The DD model provides a close fit to the viral load data in **Figure 1B** and replicates the biphasic viral load decline while excluding the dynamics of specific immune responses (**Smith et al., 2018**). Unsurprisingly, the CD8$^+$ T cell model is also capable of reproducing the biphasic viral load decay (**Figure 1B** and **Appendix 2—figure 1**). In that model, infected cell clearance is split into terms for non-specific clearance ($\delta$) and CD8$_E$-mediated clearance ($\delta_E(I_2, E) = \delta_E E/(K_\delta + I_2)$) (**Appendix 2—figure 1**).

Because the CD8$^+$ T cell model is more mechanistic than the DD model, most of the correlations between the parameters common to both models (i.e., the rates of virus infectivity ($\beta$), virus production ($p$), and virus clearance ($c$)) were reduced (**Appendix 2—figure 1A**). In addition, the correlations between the infected cell clearance parameters ($\delta_d$ and $K_\delta$ or $\delta_E$ and $K_{\delta_E}$) and between the rate of virus infectivity ($\beta$) and their ratios ($\delta_d/K_\delta$ or $\delta_E/K_{\delta_E}$) were abolished (**Figure 2—figure supplement 1–2**). There was a negative correlation between the infected cell clearance parameters ($\delta$ and $\delta_E$; **Figure 2B**), which may reflect the connection between the efficacy of early immune mechanisms and the CD8$^+$ T cell response. This result is in line with experimental evidence that the innate immune responses modulate the activation of adaptive immunity (**Iwasaki and Medzhitov, 2015**; **Luster, 2002**; **Iwasaki and Medzhitov, 2010**; **Le Bon and Tough, 2002**; **Jain and Pasare, 2017**).

The differences in model structure between the two models yielded changes in parameter sensitivity and model behavior during the rapid viral clearance phase (**Appendix 2—figure 1**). In the DD model, the most sensitive parameter is the infected cell clearance, $\delta_d$ (**Appendix 2—figure 1**). A 50% decrease in this parameter resulted in a ~7 d delay in viral resolution (**Appendix 2—figure 1**; **Smith et al., 2018**). In the CD8$^+$ T cell model, however, viral resolution is delayed by <1 d if the CD8$_E$-mediated infected cell clearance parameter ($\delta_E$) is reduced by 50% (**Appendix 2—figure 1**, **Appendix 3—figure 1**). The rates of CD8$_E$ expansion ($\eta$) and decay ($d_E$) are sensitive and, thus, significantly influence the viral resolution kinetics (**Figures 1–2**). A 50% decrease in $\eta$ results in a ~6 d delay in recovery (**Appendix 2—figure 1**, **Appendix 3—figure 1**) whereas a 48% decrease in $\eta$ prolongs the infection by ~30 d (**Figure 3D–E**). This bifurcation in recovery time is a unique feature of the CD8$^+$ T cell model.

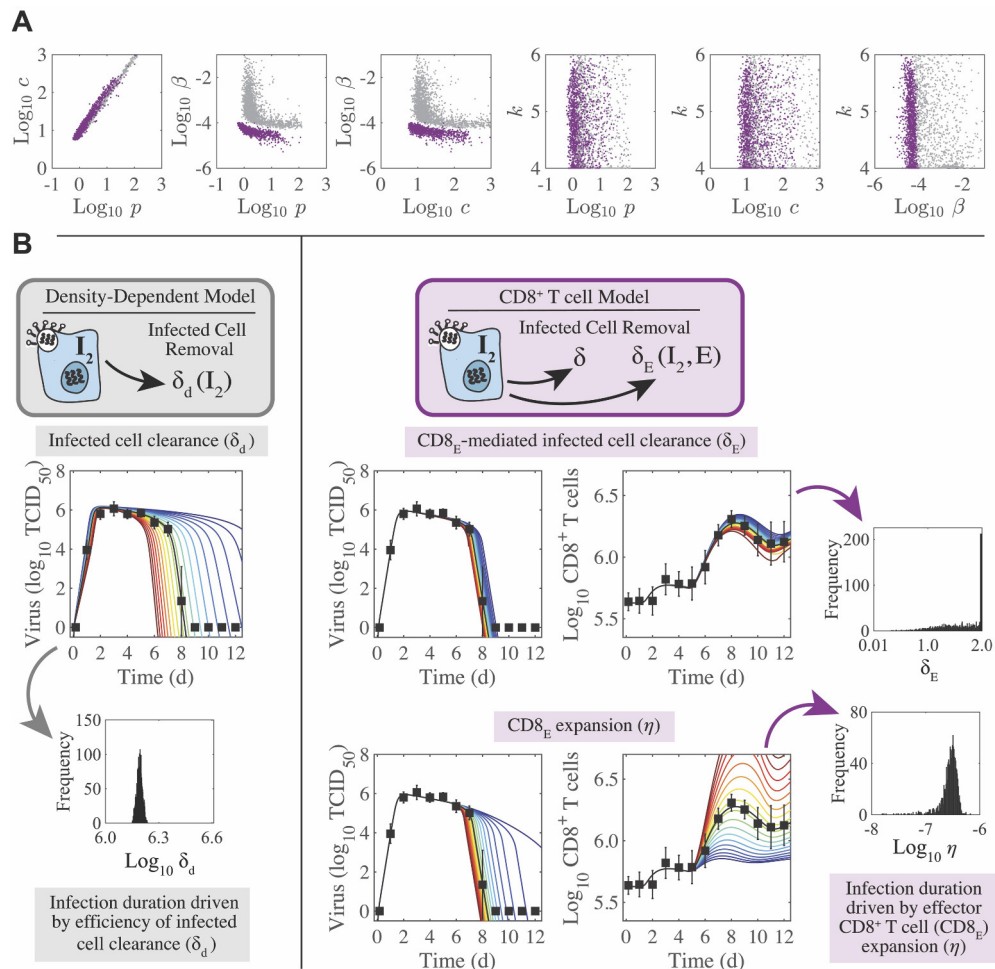

**Appendix 2—figure 1.** Parameter behavior of the density-dependent model and the CD8$^+$ T cell model. (**A**) Comparison of parameters that were common between the density-dependent model (gray, *Equation (A9)-(A12)*) and the CD8$^+$ T cell model (purple, *Equation (1)-(6)*). Correlations were evident between parameters relating to the rates of virus infectivity ($\beta$), virus production ($p$), and virus clearance ($c$). However, the strength of the correlation was significantly reduced in the CD8$^+$ T cell model. The eclipse phase parameter ($k$) was not well-defined in either model. (**B**) In the density-dependent model (gray), the viral kinetics and the infection duration were sensitive to small changes in the infected cell clearance parameter ($\delta_d$). This parameter was well-defined with a narrow 95% CI. In the CD8$^+$ T cell model (purple), changing the CD8$_E$-mediated infected cell clearance parameter ($\delta_E$) had little impact on viral kinetics or CD8$^+$ T cell kinetics. However, these kinetics were most sensitive to changes in the rate of CD8$_E$ expansion ($\eta$), which was well-defined with a narrow 95% CI.

## Appendix 3

## Regulation of the CD8$^+$ T cell Response

To further understand the regulation of the CD8$^+$ T cell response, we examined the 2-D parameter ensembles (*Figure 2A–C*, *Figure 2—figure supplements 1–2*) and the results from the sensitivity analysis (*Appendix 3—figure 1–2*). Overall, few parameters were correlated. There was an expected, although small, positive correlation between the rate of CD8$_E$ infiltration ($\xi$) and the associated half-saturation constant ($K_E$) (*Figure 2—figure supplement 1–2*), which represents the coordination between CD8$_E$ recruitment and the processes that prevent an overabundance of these cells. Likewise, a negative correlation was detected between the rate of initial CD8$_E$ influx ($\xi$) and the initial number of CD8$^+$ T cells ($\hat{E}_0$) (*Figure 2—figure supplements 1–2*). The influx rate ($\xi$) was also positively correlated with the delay in CD8$_E$ expansion ($\tau_E$) (*Figure 2—figure supplements 1–2*). The rates of CD8$_E$ expansion ($\eta$) and decay ($d_E$) are correlated (*Figure 2C*), indicating a balance between these two processes. This correlation was expected and reflects the coordination of mechanisms that regulate CD8$^+$ T cell numbers, which may be necessary to limit excessive immunopathology while still resolving the infection (*Moskophidis and Kioussis, 1998*; *Duan and Thomas, 2016*; *La Gruta et al., 2007*). Further, because of this correlation and the sensitivity of $\eta$ (*Appendix 3—figure 1*), the CD8$^+$ T cell kinetics are sensitive to changes in $d_E$ (*Appendix 3—figure 1*). However, increasing the decay rate had less impact on the viral load kinetics, comparatively. Because $d_E$ is correlated with both $\eta$ and the rate of CD8$_M$ generation ($\zeta$) (*Figure 2C*), it naturally follows that $\eta$ and $\zeta$ are correlated (*Figure 2—figure supplements 1–2*). Changing the rates of virus infectivity ($\beta$), production ($p$), or clearance ($c$) had little effect on viral load or CD8$^+$ T cell kinetics (*Appendix 3—figure 1*).

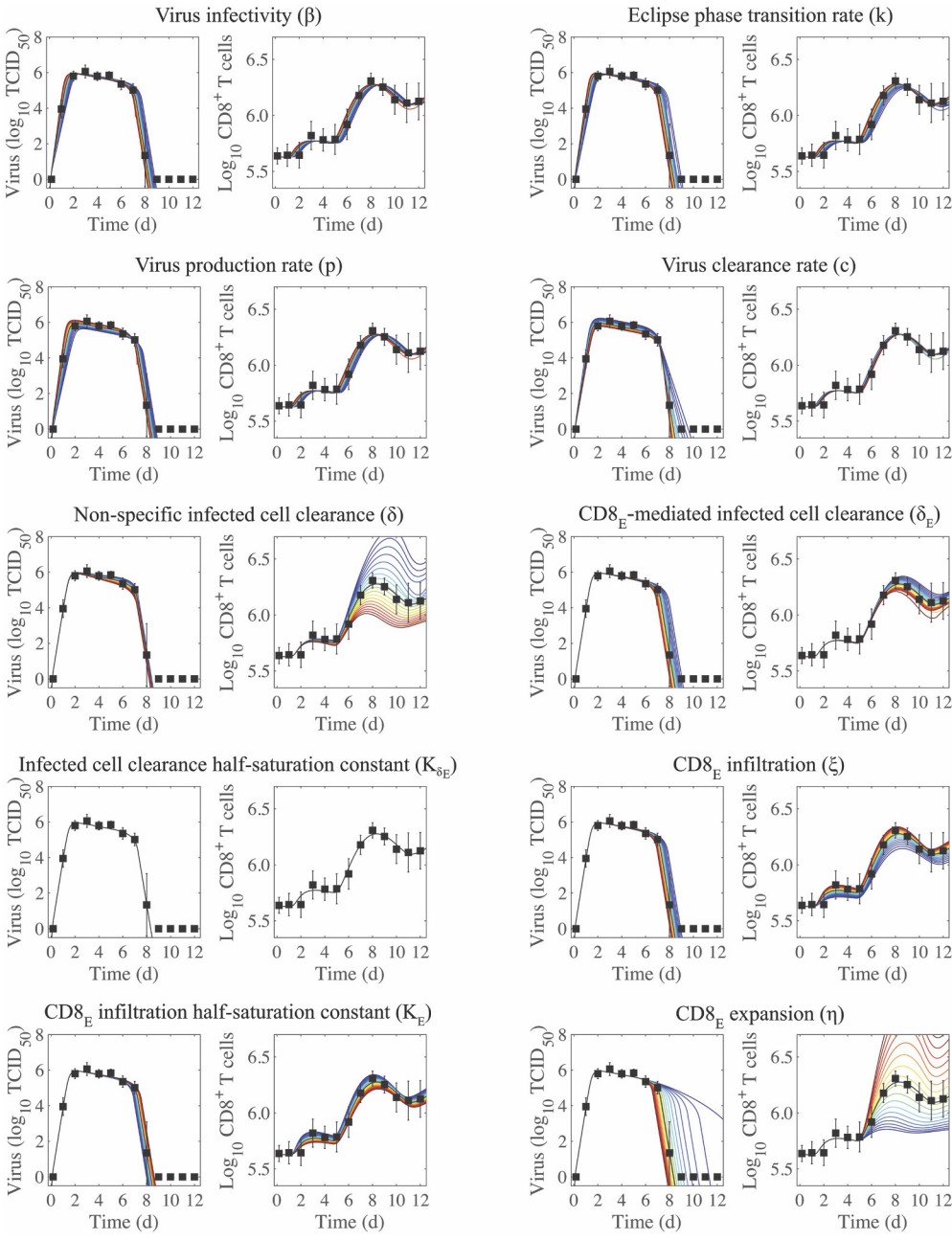

**Appendix 3—figure 1.** Sensitivity of the CD8⁺ T cell model. Solutions of the CD8⁺ T cell model (*Equation (1)-(6)*) with the indicated parameter ($\beta$, $k$, $p$, $c$, $\delta$, $\delta_E$, $K_{\delta_E}$, $\xi$, $K_E$, or $\eta$) increased (red) or decreased (blue) 50% from the best-fit value (*Table 1*). $CD8_E$ denotes effector CD8⁺ T cells.

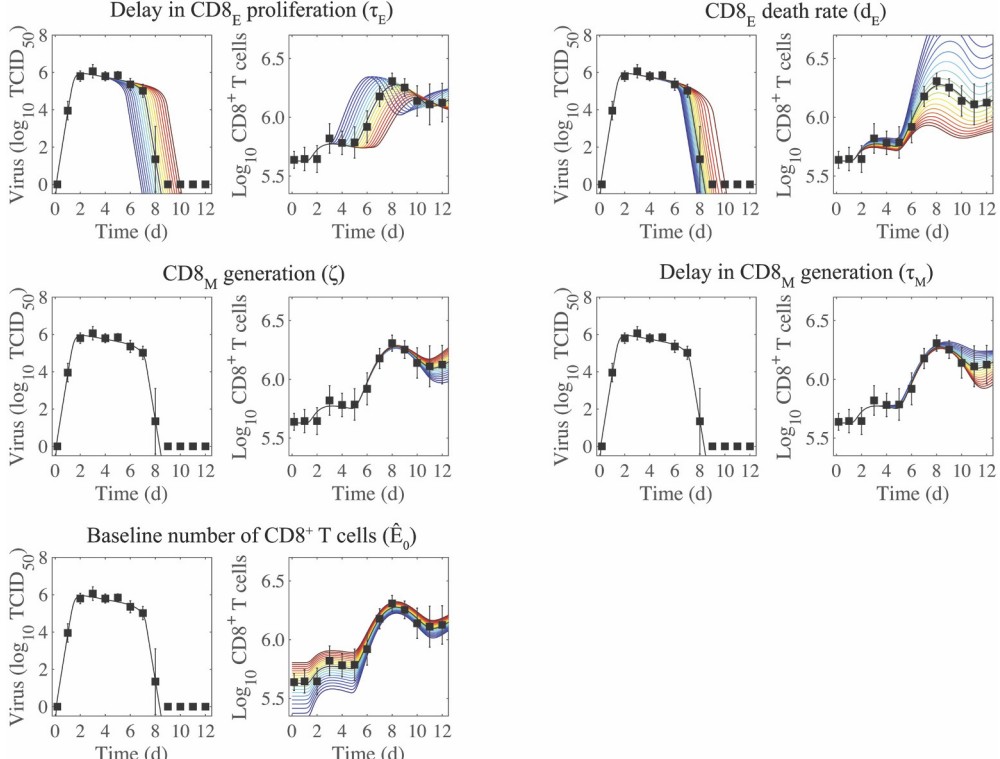

**Appendix 3—figure 2.** Sensitivity of the CD8$^+$ T cell model. Solutions of the CD8$^+$ T cell model (*Equation (1)-(6)*) with the indicated parameter ($\tau_E$, $d_E$, $\zeta$, $\tau_M$, or $\hat{E}_0$) increased (red) or decreased (blue) 50% from the best-fit value (*Table 1*). CD8$_E$ and CD8$_M$ denote effector and memory CD8$^+$ T cells, respectively.

