## [Decision Letter]

Thank you for submitting your work entitled "Dynamically linking influenza virus infection with lung injury to predict disease severity" for consideration by *eLife*. Your article has been reviewed by 3 peer reviewers, and the evaluation has been overseen by a Reviewing Editor and a Senior Editor. The following individual involved in review of your submission has agreed to reveal their identity: Joshua T Schiffer.

Our decision has been reached after consultation between the reviewers. Based on these discussions and the individual reviews below, we regret to inform you that your work will not be considered further for publication in *eLife*.

The reviewers all appreciated the attention to an important question and the combination of experimental data and modeling to understand immune control of influenza in mice. However, the decision to reject the manuscript was based on two major factors:

1) The use of total CD8 rather than antigen-specific or 'active' (cytokine producing) cells.

2) The lack of statistical analysis in comparing different (and quite complex) models (especially since many of the conclusions rested upon model comparison).

As indicated, all reviewers were excited by the approach to combining novel experimental and mathematical approaches and felt that the validity of the conclusions and priority of the manuscript would increase if these issues could be addressed. However, *ELife* requires that authors are able to complete revisions in less than two months and it was concluded that it is unlikely that these concerns could be addressed within this timeframe.

I hope that this result will not discourage your work in this area or future submission to *eLife*.

Essential revisions:

[Editors’ note: the authors submitted for reconsideration following the decision after peer review. What follows is the decision letter after the first round of review.]

*Reviewer #1:*

This is a tour de force paper describing a set of elegant experiments specifically designed for mathematical model testing and validation. The model takes the unprecedented step of linking kinetics of viral shedding, generation of infected cells, CD8^+^ T cell response and development of lung injury. Scientific conclusions are justified based on the analysis. The prediction that CD8 expansion but not levels of CD8^+^ T cells impact time to viral elimination is interesting and novel. The idea that infected cell density lowers Tcell-mediated clearance is also interesting and suggests a bottleneck to rapid immune clearance once a threshold of severity is surpassed.

Figures 3 and 4 are particularly instructive and interesting.

Overall, the paper could be substantially improved in terms of interpretation of the results and clarification of language regarding scientific conclusions. There are also a few slightly unclear sections.

1) In the introduction and conclusion, there is no mention that the balb c murine model may not capture the pathophysiology of influenza infection in humans. In particular, the relative contributions of humoral versus cell-mediated containment of infection may differ. The concepts of original antigenic sin and differential host susceptibility which may account for substantial heterogeneity in disease severity in human adults, are not captured in mice. The authors should directly admit this limitation and specifically acknowledge that the conclusions of their mathematical model (as elegant as they are) may not be fully generalizable to human infection. In addition, cited articles regarding flu pathogenesis should be labelled according to whether they are from humans, ferrets or mice. I would strongly consider reorganizing the discussion towards summarizing what conclusions might be relevant for human infections and what experimental data could be gathered to validate model predictions.

2) A key component of influenza pathogenesis is completely neglected, which is that a majority of infected people do not develop any lung disease. Instead, infection is limited to the upper airways. Therefore, a key role of the acquired immune response may be limiting spread from the upper to the lower airway. Under the best of circumstances, the author's model is relevant to only a subset of human cases. While the experimental system does not allow assessment of progression from upper to lower tract disease, this should be acknowledged as a major limitation of the system.

3) Figure 1: It is stated multiple times in the paper that Figure 1 demonstrates heterogeneity in viral loads. This is not the case as the standard deviation merely shows the confidence in the mean. Individual data points should be plotted to demonstrate heterogeneity in viral load and CD8^+^ T cell counts at each time point across mice.

4) Line 108: What percentage of infections are cleared within 4h? Are these included in the means and SDs in Figure 1? What is the proposed mechanism of these aborted infections?

5) A critical component of the model that is only mentioned in passing throughout the paper is the time delay in CD8 proliferation. The time delay parameter is as critical to the model as the density dependent CD8 killing rate (see Appendix 3 Figure A4). Yet the biology underpinning this delay is given short thrift in the discussion. It would also be useful to show that a model without this assumption fails to fit the data.

6) Table 1: are all parameter fitted? Are there references from the literature confirming that some of these values are realistic?

7) The term "model ensembles" described in line 148 is never adequately defined. The subsequent section of the paper is therefore confusing. Is this essentially describing the fact that several parameters are not identifiable because they are correlated to achieve model fit? This would not negate the model's validity but should at least be stated. Does Figure 2A only include parameter sets that resulted in optimal fit to the data (<5% error from the best fit) as described in the methods? Please clarify.

8) Line 320: the authors never show that small changes in viral load can lead to major changes in disease severity. They actually could do this with the sensitivity analysis output and it would be quite useful. However, without these simulations explicitly shown, this sentence should be removed.

9) Line 386: this is a false statement. See Figure 3 in https://www.nejm.org/doi/full/10.1056/NEJMoa1716197

*Reviewer #2:*

General assessment: The study presents a thorough combination of different experimental measurements with mathematical modeling to link viral dynamics and disease pathology in a mouse model of influenza infection. They find a very remarkable connection between the area of lung injured, the CD8^+^ T cell response and the development of disease symptoms of the animals. The discovered connections could advance the understanding of influenza virus infection in mice, but, in my view, the proposed predictive value should be further corroborated by appropriate analyses. I consider this as a very interesting study that is methodologically sound and innovative. However, I would have some comments (see below) that question the significance of this work for the field as requested by *eLife*.

1) A major concern affects the CD8^+^ T cell response used for modeling. It is stated that the total number of CD8^+^ T cells rather than the virus specific CD8^+^ T cells were used as the later often show varying dynamics (line 447-451). In which way would the use of virus-specific CD8^+^ T cells skew the results as mentioned? It would be an interesting question if the found density-dependent CD8^+^ T cell mediated clearance also holds if only virus-specific CD8^+^ T cells are considered. In addition, a population of memory cells was explicitly included in the model to reduce the number of CD8^+^ T cells responsible for clearing infected cells. This might not be necessary if virus-specific cells show a different dynamic than the total lung-resident CD8^+^ T cell population. In this regard, also the statement that the magnitude rather than the efficacy of the CD8 + T cells controls clearance could be questioned (line 98-99), as these aspects were not separately investigated in the experimental data nor the modeling framework.

2) Although the authors investigate the impact of the magnitude of the CD8^+^T cell response on the recovery time using their identified model (Figure 3), they do not use/extend these analyses to show the predicted changes for disease dynamics and pathology as based on the "workflow" shown in Figure 5. I think the study would substantially benefit if the postulated connectivity between disease dynamics and pathology, and the proposed impact of these findings on "forecasting disease progression, potential complications and therapeutic efficacy (line 23)" is shown at least theoretically for some scenarios (e.g. those used in Figure 3 that affect recovery time).

3) The analysis here benefits from the possibility to measure local viral loads and CD8^+^ T cell populations within the lungs of mice. I consider this as being rather difficult to be done in humans, as well as finding appropriate quantitative markers for disease pathology, such as weight loss. Therefore, I do not directly see the claimed ability of this study to enhance the ability to forecast disease progression and potential complications, at least not in humans. Even for mice it would be important to know if e.g. having only a limited number of measurements on the dynamics of the CD8^+^ T cell response and the viral load (e.g. until day 4 or 6 if this could be measured in vivo) is sufficient to parameterize the whole model appropriately in order to predict further dynamics.

*Reviewer #3:*

The authors present an interesting mix of modelling and experimental work looking at CD8 T cell control of influenza virus infection in mice. This extends previous work by looking at infected cell area, and coming to some slightly different conclusions on the mechanisms of T cell control. The strength of the conclusions is not well justified, and therefore the advance of previous work seems somewhat incremental.

1) Table 1 includes 21 parameters to fit CD8 and viral load. However, figure 1 suggests there are only 24 data points. This seems rather over-parameterized? This seems very evident when the authors justify the model, because for every potential inflection of each curve, they seem to add another parameter. But are all of these justified? Does adding additional parameters improve the fit (by AIC, for example)?

2) There appears no consideration of 'significance' in comparing the fit of different models. For example, the authors state (lines 127-135) that density dependence in clearance was a better fit than in CD8 expansion (as used previously), and just refer to the shape of curves on specific days. Surely something like an AIC for overall fit would be useful in comparing different models?

3) There seems a major confusion between 'total CD8' in lung, and virus-specific CD8. These seem used interchangeably. For example, line 166-168: 10% of (antigen specific) cells survive into memory, and here the authors comment they observe 17% (of peak total CD8). In the discussion (line 262 onwards) the authors discuss CD8E – referring to literature on antigen specific cells and comparing their results to these – when they have not measured this. This is justified because cells of different specificity have different kinetics. But since the total CD8 number is being used to predict the effects of antigen-specific cells (and this is the central conclusion of the manuscript) – surely some measure of what is functional (?cytokine secreting) or antigen-specific (tetramer or ICS positive) is necessary?

4) Model comparisons with data seem very vague. For example in Figure 4c the authors state (line 221-222) "the dynamics of the damaged cells of the lung correspond precisely to the dynamics of the maximum CD8". This sounds like it may be quantitative, but the figure just looks like they both peak around day 8 and decline. The CD8 increase on day 2 – whereas lesion does not increase until later. The same is true for the claim that AUC of infection matches active infection area (Figure 4B).

5) There seems relatively little explanation of the link between viral loads and lung lesions. For example, viral load peaks on day 2 and decreases thereafter. But lung lesion size is barely detectable at this time, and increases rapidly? The authors argue that AUC of virus = active area. How does this arise? Do infected cells produce a burst of virus and remain antigen positive for some time? What resolves antigen negative cells? How does lung lesion size (and %active inactive) directly relate to viral load? The authors make arguments around these issues, and figure 5 might lead one to believe there is some modelling to link all these, but there does not appear to be a mechanistic explanation?

6) The entire section "density dependent infected cell clearance" involves a lot of speculation on recovery time, CD8 thresholds etc. This seems entirely dependent on the model formulation, and average parameters (which are widely distributed). Is there any experimental evidence to support this modelling speculation?

[Editors’ note: further revisions were suggested prior to acceptance, as described below.]

Thank you for submitting your article "Dynamically Linking Influenza Virus Infection Kinetics, Lung Injury, Inflammation, and Disease Severity" for consideration by *eLife*. Your article has been reviewed by 2 peer reviewers, including Joshua T Schiffer as the Reviewing Editor and Reviewer #1, and the evaluation has been overseen by Sara Sawyer as the Senior Editor.

Essential Revisions:

1) Please include analyses in which δ_e does not approach its boundary value.

2) Please include AICs and model fits for models which are less supported by the data.

3) Please rewrite the abstract, introduction and discussion to highlight the scientific conclusions of the paper, rather than just the substantial technical achievements of the paper (fitting a model to several longitudinal data types in a complex and potentially representative model system of influenza).

4) Please frame the paper's conclusions and limitations more specifically in reference to their potential relevance for human infection. Specifically, highlight that the system employed in this paper is akin to a primary infection model rather than re-infection. Please also make an effort to partition past literature into mouse and human studies for better clarity.

*Reviewer #1 (Recommendations for the authors):*

The experimental data and modeling are highly robust. The conclusions of the paper are clearly supported by the results. The sensitivity analysis is particularly impressive and suggests a system that is highly conserved across a wide parameter space. Model validation with CD8^+^ depletion is a nice addition that leads to interesting and surprising conclusions. The figures are highly instructive and easy to read.

An area where the paper could be improved is conveying the actual scientific conclusions more clearly and precisely with more focused review of existing literature. The relevance of the paper's conclusions for human influenza could be discussed with more careful language.

First, the mechanistic conclusions of the work could be emphasized along with the methodology of the work. At present, these are completely lacking from the abstract which somewhat blandly just says that the paper describes a model which fits to data. From my perspective, currently underemphasized and novel / interesting conclusions are that:

1) CD8^+^ mediated killing becomes much more rapid on a per capita basis (40000 fold increase) when infected cells dip below several hundred cells approximately 7 days post infection.

2) There is a negative correlation between infected cell clearance by innate versus CD8^+^ mediated mechanisms, implying that poorer initial clearance of virus may result in more effective later killing by acquired immune mechanisms.

3) Even ~80% reduction in maximal CD8E+ levels could prolong infection by 10 days though delay in attaining these threshold CD8E+ levels due to experimental or in silico CD8^+^ depletion only delays viral elimination by a day.

4) Most interesting and counterintuitively, CD8^+^ depletion allows for considerable reductions in the size of lung lesions as well as inflammation scores and degree of weight loss during primary influenza infection. This result suggests that CD8^+^ T cells have the potential to create significant bystander damage in the lung.

Second, the introduction and discussion continue to not differentiate whether past experimental results are from humans or mice. It is somewhat misleading to cite mouse studies without acknowledging that these are from a model that in no way captures the totality of human infection conditions. For all animal models of human infection, the strengths of the model (ability to control experimental inputs and obtain frequent measurements) are counter-balanced by lack of realism. Humans have a complex background of immunity based on past vaccination and infection, different modes of exposure and other innumerable differences. In most human infections, the degree of lung involvement is minimal. Please stipulate in the review of existing literature which papers were done in mice versus humans. Please also frame conclusions of this paper in the discussion in terms of how it may or may not be relevant to human infection.

Third, this is a primary infection model, and this point also should be emphasized. The greatest relevance of the mouse model in the paper may be for pediatric infection in humans, rather than adults who have had multiple prior influenza exposures and possibly vaccinations. Presumably CD8^+^ responses can be expected to be more rapid with availability of a pre-existing population of tissue resident CD8^+^ T cells as would occur with re-infection. The results of CD8^+^ depletion prior to re-infection would potentially be very different (likely harmful) in a re-infection model and this should be discussed. This is mentioned in Line 467 but is given short attention elsewhere.

Line 60: stating that other studies have had limited success is rather insulting. Please rephrase and be more specific about why this study breaks new ground.

Line 81: "viral loads in the upper respiratory tract do not reflect the lower respiratory tract environment. " Please include a citation, remove or clarify that this is a possible confounding variable in the analysis.

Line 91: define lung histomorphometry. This is a fairly novel approach for most readers.

Line 101: This is a strong statement about viral load. Unless formal correlate studies have been done in humans (which they have not), I would day "may not be correlated" or remove altogether.

Line 201: involved with what? I am not sure what this sentence means.

Line 209: I would suggest denoting a separate section to the sensitivity analysis versus the parameter fitting as the fitted correlation between δ and δ_e appears separate mechanistically from the relationship between δ and viral clearance / total # of CD8E

Line 251: Please cite the clinical correlate oof this in the discussion. Immuncompromised humans often shed influenza (and SARS CoV-2) for months. See work from Jesse Bloom's group published in *eLife* on this subject.

Line 321 should this read "clear infected cells from the lung?" I am confused about what this sentence means.

Figure 5D: why are the dots yellow? Is the magenta line CD8 depleted?

Line 386: Has antiviral therapy been linked with extent of radiologic lung lesions in clinical trials. This would be a very atypical clinical trial endpoint so please be more precise with language. It is possible as previously mentioned in the paper that viral load may not predict lesion size or disease severity in humans.

Line 477: add degree of immunity from prior infections as a critical variable

*Reviewer #2 (Recommendations for the authors):*

This is a revised version of a previously reviewed article. The authors performed extensive additional analyses to address previous concerns and issues raised by the reviewers. However, there are a few additional points which, in my view, still would need some clarification:

1. As pointed out by one of the previous reviewers, the remarkable ability of the model to basically cover every change in the dynamics observed within the data (Figure 1) could suggest that the model is overfitting the data. In response, the authors mentioned that they performed robust fitting and sensitivity analyses, and also mentioned that they performed several different model attempts to reach at their final model. However, the current information provided, as e.g. in Figure 2 showing the parameter estimates, do not seem to fully support this claim. The authors state that "the majority of parameters are well-defined" with the exception of three parameters. However, also the death rate δ_E reaches the imposed boundary for fitting, which seems to be not addressed. As this parameter controls the CD8-mediated death rate and, thus, could be critical for the argument of a density-dependent death rate, I think this should be discussed/investigated in more detail. In addition, it could be very convincing if the AICs of some of the models tested that did not fit the data (i.e. reduced models as claimed within the response), are shown within the manuscript to support their claims. In addition, just as a suggestion, approaches that do not explicitly describe the mechanistic of the CD8^+^ T cell response but rather describe the measured responses by a spline function could be considered as well (see Kouyos et al. PLoS Comp Biol 2010), reducing the complexity of the model by still being able to examine the relationship between CD8^+^ T cell response and immunopathology.

2. Lung immunohistopathology is quantified based on tissue sections of individual mice. Did the authors analyze the whole (i.e. 3D) lung for regions of active/inactive lesions or only representative 2D tissue sections? In addition, how many mice/tissue sections were analyzed at each time point (e.g. Figure B/C)? It would be interesting to know how representative a 2D tissue section as shown in Figure 4A would be for the situation in a mouse, and also how representative it would be across mice. I apologize if I missed this information in the text but I think these details should be provided.

3. In Figure 5 B and D the difference between the fitted (black) and predicted (purple) relationships does not seem to be explained within the figure legend nor the text. I have difficulties to understand how these two things are related. The same holds true for Figure 5A, where there is no information on what the purple curve represents.

[Editors’ note: further revisions were suggested prior to acceptance, as described below.]

Thank you for submitting your thoroughly revised article "Dynamically Linking Influenza Virus Infection Kinetics, Lung Injury, Inflammation, and Disease Severity" for consideration by *eLife*. Your article has been reviewed by 1 peer reviewers, and the evaluation has been overseen by a Reviewing Editor and Sara Sawyer as the Senior Editor.

Essential Revisions:

We thank you for taking the effort to making excellent revisions and thoroughly addressing all reviewer comments. We apologize for not noticing this in the last revision but the Discussion section of the paper lacks a limitations paragraph and there are indeed several limitations that are not mentioned (lack of complete realism of the mouse model as a model of human infection; the fact that the weight loss equations that relate to inflammation are not "mechanistic" and therefore provide only some insight; other arms of the immune response are not studied in depth (humoral) experimentally and this system and may be important). A brief acknowledgment of these and other limitations, and possible next steps to address them, would really strengthen the impact of this paper.

---

## [Author Response]

Essential revisions:

[Editors’ note: the authors resubmitted a revised version of the paper for consideration. What follows is the authors’ response to the first round of review.]

Reviewer #1:This is a tour de force paper describing a set of elegant experiments specifically designed for mathematical model testing and validation. The model takes the unprecedented step of linking kinetics of viral shedding, generation of infected cells, CD8^+^ T cell response and development of lung injury. Scientific conclusions are justified based on the analysis. The prediction that CD8 expansion but not levels of CD8^+^ T cells impact time to viral elimination is interesting and novel. The idea that infected cell density lowers Tcell-mediated clearance is also interesting and suggests a bottleneck to rapid immune clearance once a threshold of severity is surpassed.Figures 3 and 4 are particularly instructive and interesting.Overall, the paper could be substantially improved in terms of interpretation of the results and clarification of language regarding scientific conclusions. There are also a few slightly unclear sections.

We appreciate the careful reading and kind words about our work. As noted, both above and below, we have added a significant amount of new data and included discussion (see below) to address the concerns raised.

1) In the introduction and conclusion, there is no mention that the balb c murine model may not capture the pathophysiology of influenza infection in humans. In particular, the relative contributions of humoral versus cell-mediated containment of infection may differ. The concepts of original antigenic sin and differential host susceptibility which may account for substantial heterogeneity in disease severity in human adults, are not captured in mice. The authors should directly admit this limitation and specifically acknowledge that the conclusions of their mathematical model (as elegant as they are) may not be fully generalizable to human infection. In addition, cited articles regarding flu pathogenesis should be labelled according to whether they are from humans, ferrets or mice. I would strongly consider reorganizing the discussion towards summarizing what conclusions might be relevant for human infections and what experimental data could be gathered to validate model predictions.

The reviewer is correct that we had previously limited the discussion of human data, but it is an important discussion point. It is possible that some features could vary, particularly given that humans and influenza viruses are highly heterogeneous and immune responses change with age, sex, race, various co-morbidities, immunologic history, etc. This heterogeneity is difficult to control for in the laboratory or clinic, and clinical data, in particular, is insufficient to dissect the contribution of each. Further, it is quite difficult and invasive to obtain lung measurements (e.g., virus or CD8s) from living humans, where even CT imaging is a rare event. This makes the animal model highly desirable and the only system to begin establishing these connections. Fortunately, the mouse model is well established, recapitulates many aspects of human influenza disease, including a striking similarity in lung pathology (noted in Lines 67-68 and 382), and provides a highly controlled, reproducible system. We agree that there is much more to do to prove the human connection, but this is well beyond the scope of this manuscript. Nevertheless, we believe that the model and results will be similar, although may need adaptations due to pre-existing immunity (even if heterosubtypic), and resident T cells are only beginning to be understood.

2) A key component of influenza pathogenesis is completely neglected, which is that a majority of infected people do not develop any lung disease. Instead, infection is limited to the upper airways. Therefore, a key role of the acquired immune response may be limiting spread from the upper to the lower airway. Under the best of circumstances, the author's model is relevant to only a subset of human cases. While the experimental system does not allow assessment of progression from upper to lower tract disease, this should be acknowledged as a major limitation of the system.

The reviewer is correct that some influenza viruses and in some individuals only a mild, upper respiratory tract (URT) infection is established and that our data is not relevant for that case given our focus on the lung. Our workflow would generally not be appropriate for URT viral loads (e.g., nasal washes) because these do not reflect the lung environment or indicate clinical severity. There is copious data showing that many influenza virus strains (e.g., 2009 pH1N1) and certain individuals (e.g., elderly, pregnancy, immunocompromised, etc.) are more susceptible to lower respiratory tract (LRT) infections and manifestations. Other factors that contribute to developing LRT infections include dose, volume of inhaled inoculum, lung function and size, etc.. LRT infections are important to understand as they place the highest burden on public health and are a major cause of hospitalizations and influenza-related mortality. Importantly, CT scans of hospitalized patients show strikingly similar pathological features as our data. To highlight this, we have included discussion and references (Lines 67-68 and 382) to reflect this data.

With respect to the question regarding migration from URT to LRT, this question is related yet distinct from this particular data and some clinical situations as it is not necessarily a qualifying event for a lung infection. From an experimental standpoint, we do have the ability to mimic the migration using a low volume inoculum, which initially stays predominantly in the nasopharynx. High dose is not technically possible in low volume inoculation, so we have that as an additional confounder. Our collaborators have done this experiment for parainfluenza infection (e.g., Burke et al. (2015)), which is quite similar to influenza. Notably, individual animals were able to be tracked in that system using bioluminescence, so the migration was visually documented. In short, that data showed that the viral loads in the lung lag by ~2 days and resulted in a lower viral load with smaller areas of infection. The reduced infection and the reduced weight is precisely in line with our findings here. Similar reductions in infected cells and weight loss occurred in the animals that were CD8-depleted (Figure 5). Thus, the connections and results here are unlikely to change in that scenario. We recently submitted a manuscript analyzing the parainfluenza data, which the reviewer will certainly find of interest.

3) Figure 1: It is stated multiple times in the paper that Figure 1 demonstrates heterogeneity in viral loads. This is not the case as the standard deviation merely shows the confidence in the mean. Individual data points should be plotted to demonstrate heterogeneity in viral load and CD8^+^ T cell counts at each time point across mice.

We believe the reviewer is referencing standard error of the mean (SEM) rather than standard deviation (SD). In this type of data, SEM is not appropriate and we use SD. With SD, the heterogeneity in the data is directly captured and reflected as plotted. However, the data for each individual is included in our data files and the viral loads of individual animals were previously published in Smith et al. (2018) Front Microbiol. Here, we do not include a figure of the individual data due to the inclusion of the raw data as mandated by *eLife* and the limited space for additional figures.

4) Line 108: What percentage of infections are cleared within 4h? Are these included in the means and SDs in Figure 1? What is the proposed mechanism of these aborted infections?

100% of the animals have undetectable virus at 4 h. The means and SD are included. However, this is not clearance per se, but rather indicates that cells were infected but not yet producing sufficient virus to be measured by TCID_50_.

5) A critical component of the model that is only mentioned in passing throughout the paper is the time delay in CD8 proliferation. The time delay parameter is as critical to the model as the density dependent CD8 killing rate (see Appendix 3 Figure A4). Yet the biology underpinning this delay is given short thrift in the discussion. It would also be useful to show that a model without this assumption fails to fit the data.

The reviewer is correct that the delay is important. This delay reflects the time it takes for antigen presentation, expansion in the lymph, and trafficking to/from the lymph (Lines 671-675).

6) Table 1: are all parameter fitted? Are there references from the literature confirming that some of these values are realistic?

~14 parameters are estimated using a global search algorithm (see Table 1 and Lines 686-730). Most of the parameters are unknown biologically and some encompass several processes, and thus are not available in the literature. We did compare the ones that had to biologically known entities (e.g., percent memory cells generated (see Lines 229-233)). These were in agreement, which is a good indication that our parameters are realistic.

7) The term "model ensembles" described in line 148 is never adequately defined. The subsequent section of the paper is therefore confusing. Is this essentially describing the fact that several parameters are not identifiable because they are correlated to achieve model fit? This would not negate the model's validity but should at least be stated. Does Figure 2A only include parameter sets that resulted in optimal fit to the data (<5% error from the best fit) as described in the methods? Please clarify.

Parameter ensembles or model ensembles are collections of parameters that produce solutions within a 95% confidence interval of the best-fit. This collection of solutions is the gray shading in all model fits (Figures 2 and 4) and the parameter ‘clouds’ in Figure 3 and S2-S3. Some parameters are non-identifiable (e.g., eclipse phase, *k*, and memory delay, t_m_), which is evident from visualizing the parameter ensembles and plotting the histograms (see Figures S2-S3). That is, non-identifiable parameters will return uniform-like distributions where nearly every value works equally well. Correlated parameters, on the other hand, are not considered non-identifiable. These parameters are still bounded and have well defined distributions. We have shown in our coinfection work that correlated parameters do not inhibit our ability to accurately estimate their values (see Smith (2018) Immunol Rev for further discussion). In that work, we were able to experimentally validate a correlated parameter and its 95% CI. It is quite useful to know of their correlation, however, in order to accurately interpret the results. This is why we always plot the 2D parameter ensembles.

8) Line 320: the authors never show that small changes in viral load can lead to major changes in disease severity. They actually could do this with the sensitivity analysis output and it would be quite useful. However, without these simulations explicitly shown, this sentence should be removed.

This is directly shown in the 95% confidence intervals surrounding the model predictions of the lesion data. That is, parameters within the 95% CIs simultaneously generate the viral load curve (V), the CD8 curve (E+E_M_+E*), the active lesion via CAUC(I_2_), and the inactive lesion via E/E_max_ (Figures 1 and 4). Thus, one should be able to directly see that small changes in viral loads (gray band around the V curve in Figure 1) can lead to larger changes in the percent of the lung infected (gray band around CAUC(I_2_)).

9) Line 386: this is a false statement. See Figure 3 in https://www.nejm.org/doi/full/10.1056/NEJMoa1716197

In that article, baloxavir does slightly better than oseltamivir, but neither do a great job of reducing viral loads as compared to placebo. For example, in Figure 3A, the mean is reduced but the error bars are extremely large and overlap with the placebo group. This does not indicate a robust efficacy, at least in our opinion. Nevertheless, we updated this line to read: “This is particularly important because current antivirals alleviate symptoms but do not s effectively lower viral loads”.

Reviewer #2:General assessment: The study presents a thorough combination of different experimental measurements with mathematical modeling to link viral dynamics and disease pathology in a mouse model of influenza infection. They find a very remarkable connection between the area of lung injured, the CD8^+^ T cell response and the development of disease symptoms of the animals. The discovered connections could advance the understanding of influenza virus infection in mice, but, in my view, the proposed predictive value should be further corroborated by appropriate analyses. I consider this as a very interesting study that is methodologically sound and innovative. However, I would have some comments (see below) that question the significance of this work for the field as requested by eLife.

We appreciate the reviewer’s assessment of our work and concerns. To help substantiate the claims, we have added a significant amount of new data as described both above and below.

1) A major concern affects the CD8^+^ T cell response used for modeling. It is stated that the total number of CD8^+^ T cells rather than the virus specific CD8^+^ T cells were used as the later often show varying dynamics (line 447-451). In which way would the use of virus-specific CD8^+^ T cells skew the results as mentioned? It would be an interesting question if the found density-dependent CD8^+^ T cell mediated clearance also holds if only virus-specific CD8^+^ T cells are considered. In addition, a population of memory cells was explicitly included in the model to reduce the number of CD8^+^ T cells responsible for clearing infected cells. This might not be necessary if virus-specific cells show a different dynamic than the total lung-resident CD8^+^ T cell population. In this regard, also the statement that the magnitude rather than the efficacy of the CD8 + T cells controls clearance could be questioned (line 98-99), as these aspects were not separately investigated in the experimental data nor the modeling framework.

To address any potential differences between use of the total CD8s and IAV-specific CD8s, we performed additional experiments that are described at the beginning of this document and in the manuscript (see Figure 1). In short, we detailed IAV-specific CD8s, including the effector phenotype CD25^-^CD43^+^ and ensured they were present in the lung parenchyma. Because they have similar dynamics as the total, our results were not skewed. The density-dependence was still necessary to include in the model, and only select parameters were altered due (see Table 1). Plotting the IAV-specific CD8s against the % inactive lesion also showed that the nonlinearity is even stronger with these cells (Figure S4/Supplementary file 1D). In addition, we further tested our model by depleting CD8s (Figure 3). The model fit these data and highlighted the density dependence (noted in Lines 270-271). With respect to our statement about the efficacy versus the magnitude, we do investigate this through a sensitivity analysis (Appendix 3) and now through depletion of the cells. Our findings suggest that the efficacy parameter was insensitive (see Appendix 3) and that this parameter was not changed during depletion even though the population level did change (see Table S1).

2) Although the authors investigate the impact of the magnitude of the CD8^+^T cell response on the recovery time using their identified model (Figure 3), they do not use/extend these analyses to show the predicted changes for disease dynamics and pathology as based on the "workflow" shown in Figure 5. I think the study would substantially benefit if the postulated connectivity between disease dynamics and pathology, and the proposed impact of these findings on "forecasting disease progression, potential complications and therapeutic efficacy (line 23)" is shown at least theoretically for some scenarios (e.g. those used in Figure 3 that affect recovery time).

The reviewer highlights an interesting point that was not included in our original submission. In response, we performed an additional experiment to specifically investigate the recovery time (described in detail at the beginning of this document, in the previous point, and in the manuscript text). This experiment is slightly different than our *in silico* experiment where we had altered the expansion rate. Isolating and changing rates in vivo is extraordinarily difficult, but altering the population levels is relatively simple. In short, we depleted CD8s during the infection and showed how virus, CD8s (Figure 3F), and weight loss (Figure 5A) change. Because viral loads were lower by 2 d π and because CD8 depletion antibodies are known to alter various aspects of the response, we refit the model to identify potential differences in model parameters (see Figure 3 and Table S1). Our analysis suggested 3 differences: lower T_0_, lower x, and higher h. Plotting the relation between the activated CD8s and virus also suggested a lower x and higher h (see Figure 3G). We chose to not perform the histormorphometry for this experiment to verify the lower T_0_ because it would have doubled the number of animals, which was not ethically justifiable. Instead, we used the opportunity to test our workflow by predicting the changes in lung lesions and inflammation (Figure 4). Then, we tested the correlation between these and weight loss (Figure 5). The primary Hill function parameters (*n*’s and *K*’s) were unaltered and the maximum (l_max_’s) was obtained directly from the estimated lesion and inflammation values. These results show that the workflow is robust, and indirectly validate the effect of depletion on T_0_.

3) The analysis here benefits from the possibility to measure local viral loads and CD8^+^ T cell populations within the lungs of mice. I consider this as being rather difficult to be done in humans, as well as finding appropriate quantitative markers for disease pathology, such as weight loss. Therefore, I do not directly see the claimed ability of this study to enhance the ability to forecast disease progression and potential complications, at least not in humans. Even for mice it would be important to know if e.g. having only a limited number of measurements on the dynamics of the CD8^+^ T cell response and the viral load (e.g. until day 4 or 6 if this could be measured in vivo) is sufficient to parameterize the whole model appropriately in order to predict further dynamics.

The reviewer raises interesting questions and is correct that measuring viral loads or CD8s from the human lung is quite challenging. CD8s can be measured from the blood, but these don’t directly reflect the lung environment. Although influenza does replicate well in the upper respiratory tract (URT), which is easily accessible, URT measurements are limited in their utility. This is because similar viral loads can be observed in patients with different outcomes due to the URT being an initial contact site. In some cases, patients have x-rays or CT scans performed. CT scans look strikingly similar to our lung histomorphometry (noted in Lines 67-68 and 382), which gives us hope that our findings here would be translatable to humans. Another challenge will be finding a symptom that relates in the same way as weight loss. Humans experience a range of symptoms (e.g., fever, cough, shortness of breath, fatigue, etc.), which do not necessarily have the same timescales or act as the best predictor of disease (noted in Lines 104-106). Measurement of these can also be quite subjective. These difficulties highlight the importance of animal and quantitative studies like ours to begin establishing these dynamics and connections.

The question about whether less data could be used is interesting. Our CD8 depletion data offered us the possibility to analyze the data with fewer time points (5 time points instead of 13) and fewer measurements (no lesions or inflammation scoring). Fewer time points does reduce the number of parameters that one would be able to estimate with statistical confidence. Nevertheless, the results showed that our workflow could still be used. Further, had the depletion antibody not been known to affect the rates of the CD8 response, we likely could have inferred their dynamics without data. Nevertheless, these new data and analyses do support the utility of less data.

Reviewer #3:The authors present an interesting mix of modelling and experimental work looking at CD8 T cell control of influenza virus infection in mice. This extends previous work by looking at infected cell area, and coming to some slightly different conclusions on the mechanisms of T cell control. The strength of the conclusions is not well justified, and therefore the advance of previous work seems somewhat incremental.

We have updated our manuscript in numerous ways to better highlight the novelty and importance of this work. As noted above, we added a significant amount of data and analyses that further increase the novelty and substantiate the model, analyses, and findings. The reviewer is correct that we did follow up our own work on a simplified density-dependent model (Smith et al. (2018) Front Microbiol). We described this and highlighted that we came to the same conclusion with respect to the density-dependent clearance (see Lines 42, 167, 399-408, and Appendix 2). In addition, we added further insight into this term and how to appropriate interpret the smaller model (see Appendix 2). As the reviewer mentioned, this conclusion is distinct from some models, which assume the density-dependence is in the effector expansion. We investigated those models and clearly show that these other models do not fit our data (Appendix 1).

While this small portion of our work could be considered as incremental, there are numerous other aspects that significantly set our work apart from others with respect to both data, modeling, and insight. These include (1) quantitative data and model fits to various phenotypes of CD8s, (2) validated CD8 model and findings, (3) validated infected cell dynamics, (4) quantitative data and modeling of CD8 depletion, (5) quantitative data and modeling of lung injury, (6) quantitative data and modeling of inflammation, and (7) connection of both pathologic measurements with weight loss along with validation. Some studies, including some of our own, have attempted to predict immunopathology or damage, often using these interchangeably, but our data highlight their inaccuracies and clearly show that these are distinct. We are aware of a two studies that did attempt to follow up on our own novel suggestion to use weight loss data (see Smith and Perelson (2011) WIRES Sys Biol). However, one had (self-proclaimed) failure (Price et al. (2015) J Theor Biol) and another did not model any other feature to substantiate their equation for symptom scores (Manchanda et al. (2014) Biosystems). These are discussed in Lines 57-60 and 426-428. If there are other papers we have inadvertently missed that were published prior to the publication of our preprint (Feb 2019), we would be happy to include those. We hope this explanation and our new data and analyses more clearly show the numerous novelties and importance of this work.

1) Table 1 includes 21 parameters to fit CD8 and viral load. However, figure 1 suggests there are only 24 data points. This seems rather over-parameterized? This seems very evident when the authors justify the model, because for every potential inflection of each curve, they seem to add another parameter. But are all of these justified? Does adding additional parameters improve the fit (by AIC, for example)?

In Figure 1B, there are a total of 260 data points across 13 time points and 2 populations (virus and CD8 T cells). There are 14 estimated parameters in Table 1. Thus, the model is not overparameterized. Of note, the general rule of 1 parameter per 1 time point does not apply when multiple variables are involved, so numerous additional parameters could have also been fit without concern of overfitting. We’re unsure what is meant by “for every potential inflection of each curve, they seem to add another parameter”. We do not choose parameters based on inflection points. We build our models with extreme care and with the minimum number of terms and parameters needed to fit and explain the data. Our general model building algorithm is to develop it term-by-term while simultaneously testing various functional forms, beginning always with linear terms then moving to nonlinear terms if necessary. At the end, we have typically gone through dozens to hundreds of different model formulations. We do not include these in a manuscript because that would be extremely cumbersome and unnecessary. In addition, we perform robust fitting and sensitivity analyses (Figures 1-3, A2-A4) so that we and our readers have the knowledge needed to appropriately interpret the model and its results.

2) There appears no consideration of 'significance' in comparing the fit of different models. For example, the authors state (lines 127-135) that density dependence in clearance was a better fit than in CD8 expansion (as used previously), and just refer to the shape of curves on specific days. Surely something like an AIC for overall fit would be useful in comparing different models?

We do consider significance when evaluating models but only do so when it is statistically appropriate. We did not initially consider them when evaluating alternate models because it is not technically sound to compare model fits that use different data. Nevertheless, in response to the reviewer, we did do this by first including all data even though the alternate model was not fit to d3-5 and then excluding d3-5 even though our model was fit to these. We further examined how the models faired against the lung-specific CD8 data. In all cases, the AICs of the alternate models were statistically worse and could not be supported. We now include this analysis in Table S2/Supplementary file 1G along with a direct visual comparison of the alternate models in Figure A1.

3) There seems a major confusion between 'total CD8' in lung, and virus-specific CD8. These seem used interchangeably. For example, line 166-168: 10% of (antigen specific) cells survive into memory, and here the authors comment they observe 17% (of peak total CD8). In the discussion (line 262 onwards) the authors discuss CD8E – referring to literature on antigen specific cells and comparing their results to these – when they have not measured this. This is justified because cells of different specificity have different kinetics. But since the total CD8 number is being used to predict the effects of antigen-specific cells (and this is the central conclusion of the manuscript) – surely some measure of what is functional (?cytokine secreting) or antigen-specific (tetramer or ICS positive) is necessary?

We apologize for the confusion. To make this more clear, we added several new pieces of data and model fits. We’ll keep our response here brief, but a more detailed description in included at the beginning of this document and in the manuscript. We added data that define the difference between the total CD8, lung-specific CD8, and specific effector and memory phenotypes (Figures 1, S1). These data support our original approach, showed that our model predicted dynamics for CD8_E_ were reflective of CD8s in the lung parenchyma, and showed how parameters change if we assume only certain phenotypes are in action (Table 1).

4) Model comparisons with data seem very vague. For example in Figure 4c the authors state (line 221-222) "the dynamics of the damaged cells of the lung correspond precisely to the dynamics of the maximum CD8". This sounds like it may be quantitative, but the figure just looks like they both peak around day 8 and decline. The CD8 increase on day 2 – whereas lesion does not increase until later. The same is true for the claim that AUC of infection matches active infection area (Figure 4B).

Figure 4 is not a model fit but rather an overlay of the model prediction and the data. We specifically use the verbiage “correspond” and “match” within the text to indicate that it is an overlay of the model and data rather than a fit. To make this more clear, we added the words “fit” and “predicted” to most figures so that the reader can clearly distinguish these. Figure 4, specifically, shows that they correspond and that the model accurately predicts these lung pathologic features. It should be clear that the active lesion is the CAUC rather than AUC. In addition, it is not simply the CD8s, which do peak at day 8, but rather the relative CD8s. To more clearly illustrate the nonlinear relation between CD8s and the inactive lesion, we also added Figure S4F:

5) There seems relatively little explanation of the link between viral loads and lung lesions. For example, viral load peaks on day 2 and decreases thereafter. But lung lesion size is barely detectable at this time, and increases rapidly? The authors argue that AUC of virus = active area. How does this arise? Do infected cells produce a burst of virus and remain antigen positive for some time? What resolves antigen negative cells? How does lung lesion size (and %active inactive) directly relate to viral load? The authors make arguments around these issues, and figure 5 might lead one to believe there is some modelling to link all these, but there does not appear to be a mechanistic explanation?

We apologize for the confusion regarding our data. The technical description of histomorphometry is provided in the Methods (Lines 622-638). We included more detail below to further explain type of data and experimental procedures:

In our histomorphometric staining, the cells are stained with an antibody against the influenza nucleoprotein (abbreviated as NP). The labeled NP+ cells are infected cells, which should not be confused with full virions or viral loads. Infected cells express all influenza proteins, so they are able to be labeled. NP is expressed in abundance and thus widely used for staining. Virions do not express internal proteins like NP on their surface, so these are not stained in the process. Further, the influenza virus is an enveloped virus, which means that the virus requires an intact, living cell to produce virions (no bursting). Viral loads are a measure of extracellular infectious virus, and we quantify them using TCID_50_. Because of these differences, we never claim that “AUC of virus = active area”. Rather, it is the CAUC, which is distinct from AUC, of productively infected cells (CAUC(I_2_)) because histomorphometry is a measurement of the infected cell area. Obviously, free virus is what causes a cell to be infected (one could consider this the “mechanism”), so virus is indirectly related to the histomorphometry measurement. The indirect nature is noted in Figure 6 by an arrow from virus to the model (to estimate the infected cell dynamics) and from the model to the histomorphometry. “Inactive” lesions are areas that were previously infected. Once a cell becomes infected, it will either die from infection or be removed by an immune cell (one could consider this the “mechanism”). Both the active and inactive areas are defined and quantified by a board-certified pathologist who is blinded from the study.

6) The entire section "density dependent infected cell clearance" involves a lot of speculation on recovery time, CD8 thresholds etc. This seems entirely dependent on the model formulation, and average parameters (which are widely distributed). Is there any experimental evidence to support this modelling speculation?

There is prior biological evidence that the rate is density-dependent, which is discussed on Lines 49-52. We did enhance this discussion and included additional references (within Lines 409-431). In addition, our analysis in Figure S4F (noted above), our data and model fit to the CD8 depletion data (Figure 3), the close comparison of our model with the lung pathology (Figure 4), and the lack of a robust fit when this density dependence is excluded (Appendix 1) provide further support. In our opinion, this is quite a bit of supportive evidence and we do not have reason to believe that it would be dependent on model formulation. Where one would need to be cautious is only fitting the CD8 peak and expecting the model to be correct.

[Editors’ note: what follows is the authors’ response to the second round of review.]

Essential Revisions:1) Please include analyses in which δ_e does not approach its boundary value.

Below is the ensemble plot of d_E_ and d with larger bounds on d_E_ (no other parameters are significantly altered). This shows that d approaches zero with increasing values of d_E_. Mathematically, this is due to the viral loads during the first decay phase (days 2-6) having little dynamics and thus being insufficient to distinguish between two mechanisms of infected cell clearance. Biologically, it would suggest that CD8s are the only mechanism of infected cell clearance throughout the entire infection, which we believe is incorrect. Thus, in restricting d_E_, we are able to maintain biological relevancy. In addition, this allows us to keep our results consistent with prior publications where d has been shown to dictate the slope in the first decay phase. That is, the viral load slope is -0.2 log_10_ TCID_50_/day (Smith et al. (2018) Front Microbiol), which equates to d=0.46 (i.e., V(t)=e^-dt^ as derived in Smith et al. (2010) J Math Biol) in the absence of other mechanisms. Generally speaking, restricting parameters to biologically meaningful ranges is a relatively common practice (e.g., k is a prime example where numerous papers have either fixed or restricted the range of this parameter). We did not include this figure in our revised version as we feel its inclusion does not add value to the manuscript, but would be happy to if the Editors or Reviewer think it would be beneficial. We did add a brief explanation about the parameter restriction in Lines 750-752.

**Author response image 1. sa2fig1:** 

2) Please include AICs and model fits for models which are less supported by the data.

Fits and AICs were previously included in Appendix-Figure 1 and Supplementary file 1 (previously Supplementary file 1G; tables are not supported in *eLife*’s appendix box within LaTeX). These are referenced in Lines 194, 496, 691, 695, and 699.

3) Please rewrite the abstract, introduction and discussion to highlight the scientific conclusions of the paper, rather than just the substantial technical achievements of the paper (fitting a model to several longitudinal data types in a complex and potentially representative model system of influenza).

We updated the abstract, the last paragraph of the introduction (Lines 130-147), and portions of the Discussion (Lines 401-406, 414-416, 423-427, 446-450, 500-502) to better highlight our results.

4) Please frame the paper's conclusions and limitations more specifically in reference to their potential relevance for human infection. Specifically, highlight that the system employed in this paper is akin to a primary infection model rather than re-infection. Please also make an effort to partition past literature into mouse and human studies for better clarity.

Thank you for the suggestions. We updated the text to better highlight the human relevance and distinguish results between different host species in Lines 43, 61, 69, 74, 77, 88, 99, 101, 111, 116, 118, 120, 400-406, 410-411, 413, 435, 437, 446-450, 452, 463, 471, 485, and 846. We use the words ‘human’, ‘clinical’, and ‘patient’ to denote humans and the words ‘animal’, ‘murine’, and ‘experimental’ to denote animal models. Further distinguishment in some areas comes from the methods noted (e.g., CT scans/imaging is used/relevant for humans but not animals (this would be a microCT)). Because many features of the infection are observed in both humans and animals (see, for example, 10.1016/j.jim.2014.03.023 and 10.3390/pathogens3040845), underscoring the strong relevance of animal models to study influenza (noted in Lines 453-454), we limited specifying this in every sentence as we feel it would reduce the readability. In addition, we feel that splitting up the references mid-sentence would also reduce readability given the journal’s reference style, and have left most at the end of a sentence. The references themselves should provide a reader to easily distinguish.

Reviewer #1 (Recommendations for the authors):The experimental data and modeling are highly robust. The conclusions of the paper are clearly supported by the results. The sensitivity analysis is particularly impressive and suggests a system that is highly conserved across a wide parameter space. Model validation with CD8^+^ depletion is a nice addition that leads to interesting and surprising conclusions. The figures are highly instructive and easy to read.An area where the paper could be improved is conveying the actual scientific conclusions more clearly and precisely with more focused review of existing literature. The relevance of the paper's conclusions for human influenza could be discussed with more careful language.

Thank you for the suggestions. As mentioned above, we updated the abstract and the text to better highlight the biological conclusions in addition to the mathematical conclusions. We also included some additional explanation on the specific points below.

First, the mechanistic conclusions of the work could be emphasized along with the methodology of the work. At present, these are completely lacking from the abstract which somewhat blandly just says that the paper describes a model which fits to data. From my perspective, currently underemphasized and novel / interesting conclusions are that:1) CD8^+^ mediated killing becomes much more rapid on a per capita basis (40000 fold increase) when infected cells dip below several hundred cells approximately 7 days post infection.2) There is a negative correlation between infected cell clearance by innate versus CD8^+^ mediated mechanisms, implying that poorer initial clearance of virus may result in more effective later killing by acquired immune mechanisms.3) Even ~80% reduction in maximal CD8E+ levels could prolong infection by 10 days though delay in attaining these threshold CD8E+ levels due to experimental or in silico CD8^+^ depletion only delays viral elimination by a day.

In our CD8 depletion data, the entire infection is altered (see Lines 270-303). That is, our results suggest that there are fewer infected cells initially infected, which leads to lower viral loads at d2 and will automatically result in fewer CD8s. However, the depletion antibody itself is known to directly alter CD8s (see Lines 282-284), so one cannot make direct, quantified conclusions about the precise reduction percentage under this experimental condition.

4) Most interesting and counterintuitively, CD8^+^ depletion allows for considerable reductions in the size of lung lesions as well as inflammation scores and degree of weight loss during primary influenza infection. This result suggests that CD8^+^ T cells have the potential to create significant bystander damage in the lung.

While bystander damage may be present, our CD8 depletion data do not directly show bystander damage. As we mention above, one important aspect of depleting CD8s prior to infection was that it reduced the viral loads early on. We believe this is likely a result of immune activation from the initial CD8 kill-off (noted in Lines 284-285). A decrease in target/infected cells will automatically reduce the number of total cells that become infected and, thus, reduce the lesioned area of the lung. This was verified by the reduced weight loss, and no modifications were made to the predictions (Figure 4) or correlation to weight loss (Figure 5).

Second, the introduction and discussion continue to not differentiate whether past experimental results are from humans or mice. It is somewhat misleading to cite mouse studies without acknowledging that these are from a model that in no way captures the totality of human infection conditions. For all animal models of human infection, the strengths of the model (ability to control experimental inputs and obtain frequent measurements) are counter-balanced by lack of realism. Humans have a complex background of immunity based on past vaccination and infection, different modes of exposure and other innumerable differences. In most human infections, the degree of lung involvement is minimal. Please stipulate in the review of existing literature which papers were done in mice versus humans. Please also frame conclusions of this paper in the discussion in terms of how it may or may not be relevant to human infection.

As mentioned above, we updated the text to better highlight the human relevance and distinguish results between different host species in Lines 43, 61, 69, 74, 77, 88, 99, 101, 111, 116, 118, 120, 400-406, 410-411, 413, 435, 437, 446-450, 452, 463, 471, 485, and 846. We use the words ‘human’, ‘clinical’, and ‘patient’ to denote humans and the words ‘animal’, ‘murine’, and ‘experimental’ to denote animal models. Further distinguishment in some areas comes from the methods noted (e.g., CT scans/imaging is used/relevant for humans but not animals (this would be a microCT)). Because many features of the infection are observed in both humans and animals (see, for example, 10.1016/j.jim.2014.03.023 and 10.3390/pathogens3040845), underscoring the strong relevance of animal models to study influenza (noted in Lines 453-454), we limited specifying this in every sentence as we feel it would reduce the readability. In addition, we feel that splitting up the references mid-sentence would also reduce readability given the journal’s reference style, and have left most at the end of a sentence. The references themselves should provide a reader to easily distinguish.

Third, this is a primary infection model, and this point also should be emphasized. The greatest relevance of the mouse model in the paper may be for pediatric infection in humans, rather than adults who have had multiple prior influenza exposures and possibly vaccinations. Presumably CD8^+^ responses can be expected to be more rapid with availability of a pre-existing population of tissue resident CD8^+^ T cells as would occur with re-infection. The results of CD8^+^ depletion prior to re-infection would potentially be very different (likely harmful) in a re-infection model and this should be discussed. This is mentioned in Line 467 but is given short attention elsewhere.

We added text to highlight that we are studying a primary infection in Lines 46, 130, and 176-177. In general, primary infections may not necessarily always equate to children as any novel strain or strain novel to that individual may act as a primary infection. It is also feasible that waning immunity would appear similar to the dynamics of a primary infection.

Because CD8 depletion would also deplete out resident and other T cells (>99% efficiency as noted in Line 273), one would expect the exact same results as we showed here. Thus, it would not be the appropriate experimental design to study recall responses. As we mentioned in our prior response, we might expect some adjustments to account primed responses and added text that highlights this in Lines 424-427 and 498.

Line 60: stating that other studies have had limited success is rather insulting. Please rephrase and be more specific about why this study breaks new ground.

We did not mean to be insulting and reworded Lines 66-67 to “…but have had not yet found the appropriate mathematical relation with the available data.” Of note, even the authors of Price et al. noted their inability to capture the dynamics: “Some trajectories, notably activated macrophages and epithelial damage are not well captured by the model, suggesting that the immunophenotype we selected for active macrophages may not be accurate, and that using animals’ weight as a proxy for epithelial damage may not be appropriate”. The novelty of our study and the gaps in the field are stated throughout the introduction and discussion.

Line 81: "viral loads in the upper respiratory tract do not reflect the lower respiratory tract environment. " Please include a citation, remove or clarify that this is a possible confounding variable in the analysis.

We’ve added several citations from both human and animal studies to Lines 90-91.

Line 91: define lung histomorphometry. This is a fairly novel approach for most readers.

This was defined in our prior revision and remains in Lines 102-104.

Line 101: This is a strong statement about viral load. Unless formal correlate studies have been done in humans (which they have not), I would day "may not be correlated" or remove altogether.

We updated Line 110 to “…may not be directly correlated…”.

Line 201: involved with what? I am not sure what this sentence means.

We were referencing effector-mediated killing and memory generation as noted earlier in the sentence. We updated Lines 210-212 to clarify: “One benefit of using the total CD8^+^ T cells is that the model automatically deduces the dynamics of effector-mediated killing and memory generation without needing to specify which phenotypes might be involved in these processes as they may be dynamically changing”

Line 209: I would suggest denoting a separate section to the sensitivity analysis versus the parameter fitting as the fitted correlation between δ and δ_e appears separate mechanistically from the relationship between δ and viral clearance / total # of CD8E

We appreciate the suggestion but have chosen to leave this part of the text unaltered.

Line 251: Please cite the clinical correlate oof this in the discussion. Immuncompromised humans often shed influenza (and SARS CoV-2) for months. See work from Jesse Bloom's group published in eLife on this subject.

The suggested article has been added to Line 474.

Line 321 should this read "clear infected cells from the lung?" I am confused about what this sentence means.

Line 332 has been updated to “…of the infected areas within the lung”.

Figure 5D: why are the dots yellow? Is the magenta line CD8 depleted?

We inadvertently left off the explanation, but added clarification to the caption and text (Lines 371374 and 380-384). The yellow markers are interstitial inflammation while the white markers are alveolar inflammation. The magenta markers/lines are the CD8 depletion prediction.

Line 386: Has antiviral therapy been linked with extent of radiologic lung lesions in clinical trials. This would be a very atypical clinical trial endpoint so please be more precise with language. It is possible as previously mentioned in the paper that viral load may not predict lesion size or disease severity in humans.

To our knowledge, CT images are not typically taken in many contexts for a variety of clinical reasons (e.g., cost, exposure to the patient, etc.), but antivirals have been linked to reductions in disease severity (e.g., see 10.1056/NEJMoa1716197). In that particular line, we mention that minor reductions in viral load are paired with more significant reductions in disease/symptom, which is reported in the referenced clinical and experimental data.

Line 477: add degree of immunity from prior infections as a critical variable

This has been added to Line 478.

Reviewer #2 (Recommendations for the authors):This is a revised version of a previously reviewed article. The authors performed extensive additional analyses to address previous concerns and issues raised by the reviewers. However, there are a few additional points which, in my view, still would need some clarification:1. As pointed out by one of the previous reviewers, the remarkable ability of the model to basically cover every change in the dynamics observed within the data (Figure 1) could suggest that the model is overfitting the data. In response, the authors mentioned that they performed robust fitting and sensitivity analyses, and also mentioned that they performed several different model attempts to reach at their final model. However, the current information provided, as e.g. in Figure 2 showing the parameter estimates, do not seem to fully support this claim. The authors state that "the majority of parameters are well-defined" with the exception of three parameters. However, also the death rate δ_E reaches the imposed boundary for fitting, which seems to be not addressed. As this parameter controls the CD8-mediated death rate and, thus, could be critical for the argument of a density-dependent death rate, I think this should be discussed/investigated in more detail.

We appreciate the reviewer noting the great fits, which are the accumulation of several years of model development and parameter estimation. As we mentioned in our previous response to the reviewer’s concern about overfitting, we have copious data to fit the model without concern for overfitting. We believe the reviewer may be confusing over-/under-fitting with parameter identifiability and parameter correlations. These three parameter estimation features are not equivalent nor do the latter two equate to any issue in fitting or in making robust conclusions. Being fortunate enough to experimentally validate our models, we have previously shown that accurate conclusions and parameter estimates can be made in the face of parameter correlations (see our prior body of work on influenza-pneumococcal infections, which is summarized in Smith (2018) Immunol Rev). Again here, we have successfully validated our model with several pieces of data. Generally speaking, nearly every model-data pairing will result in correlated parameters and some parameters that are non-identifiable. While we could fix non-identifiable parameters to a chosen value, we would obtain the same results as if we fit them due to their non-identifiability (i.e., every values works equally well). We prefer to let these parameters fit for times when this turns out to not be the case.

With respect to d_E_, this parameter is only correlated to d as one might expect (i.e., both are infected cell clearance parameters). As we noted at the beginning of this document, we have previously shown that d can be directly estimated from the slope of the viral load data during the first decay phase (see Smith et al. (2010) J Math Biol for the original derivation). Allowing larger values of d_E_ results in d approaching zero and would suggest that the CD8s are the only mechanism of infected cell clearance, which we believe to be untrue. We included a short explanation about the parameter restriction is in Lines 750-752.

In addition, it could be very convincing if the AICs of some of the models tested that did not fit the data (i.e. reduced models as claimed within the response), are shown within the manuscript to support their claims.

Fits and AICs were previously included in Appendix-Figure 1 and Supplementary file 1 (previously Supplementary file 1G; tables are not supported in *eLife*’s appendix box within LaTeX). These are referenced in Lines 194, 496, 691, 695, and 699.

In addition, just as a suggestion, approaches that do not explicitly describe the mechanistic of the CD8^+^ T cell response but rather describe the measured responses by a spline function could be considered as well (see Kouyos et al. PLoS Comp Biol 2010), reducing the complexity of the model by still being able to examine the relationship between CD8^+^ T cell response and immunopathology.

We appreciate the suggestion, and the reviewer is correct that splines are one possibility. However, we find these to be inferior due to their non-mechanistic structure, and have chosen to not explore this path because our model is quite simplistic while simultaneously being mechanistic and accurate.

2. Lung immunohistopathology is quantified based on tissue sections of individual mice. Did the authors analyze the whole (i.e. 3D) lung for regions of active/inactive lesions or only representative 2D tissue sections? In addition, how many mice/tissue sections were analyzed at each time point (e.g. Figure B/C)? It would be interesting to know how representative a 2D tissue section as shown in Figure 4A would be for the situation in a mouse, and also how representative it would be across mice. I apologize if I missed this information in the text but I think these details should be provided.

We analyzed 2D tissue sections (Line 660) from 5 animals (Lines 596 and 668). The choice of image per group was provided by the pathologist, who is blinded from the study, as a representative image. Other animals may have lesions located in different parts of the lung (as the exact location of cells becoming infected will happen by chance), but the percent infected areas are relatively similar. The error bars on the quantified data in Figure 4 can be directly used to infer how these change between animals. The complete experimental details are in Lines 653669.

3. In Figure 5 B and D the difference between the fitted (black) and predicted (purple) relationships does not seem to be explained within the figure legend nor the text. I have difficulties to understand how these two things are related. The same holds true for Figure 5A, where there is no information on what the purple curve represents.

Our apologies that we inadvertently left the explanation out of the caption and the text. Figures 5B, 5D are now explained in the caption and text (Lines 371-374 and 380-384).

[Editors’ note: what follows is the authors’ response to the second round of review.]

Essential Revisions (for the authors):We thank you for taking the effort to making excellent revisions and thoroughly addressing all reviewer comments. We apologize for not noticing this in the last revision but the Discussion section of the paper lacks a limitations paragraph and there are indeed several limitations that are not mentioned (lack of complete realism of the mouse model as a model of human infection; the fact that the weight loss equations that relate to inflammation are not "mechanistic" and therefore provide only some insight; other arms of the immune response are not studied in depth (humoral) experimentally and this system and may be important). A brief acknowledgment of these and other limitations, and possible next steps to address them, would really strengthen the impact of this paper.

We added discussion points to Lines 407-408, 446-447, 450-451, and 456. Many limitations and potential future studies were previously acknowledged in Lines 426-429, 442-446, 447-450, 451-455, 484-487, 502-503, 520-522, 539-540, 547-549, and 552-555. We apologize that we did not point these out more vehemently in our prior revision.